# Truncated Marginal Neural Ratio Estimation

**Benjamin Kurt Miller**
University of Amsterdam
b.k.miller@uva.nl

**Alex Cole**
University of Amsterdam
a.e.cole@uva.nl

**Patrick Forré**
University of Amsterdam
p.d.forre@uva.nl

**Gilles Louppe**
University of Liège
g.louppe@uliege.be

**Christoph Weniger**
University of Amsterdam
c.weniger@uva.nl

## Abstract

Parametric stochastic simulators are ubiquitous in science, often featuring high-dimensional input parameters and/or an intractable likelihood. Performing Bayesian parameter inference in this context can be challenging. We present a neural simulation-based inference algorithm which simultaneously offers simulation efficiency and fast empirical posterior testability, which is unique among modern algorithms. Our approach is simulation efficient by simultaneously estimating low-dimensional marginal posteriors instead of the joint posterior and by proposing simulations targeted to an observation of interest via a prior suitably truncated by an indicator function. Furthermore, by estimating a locally amortized posterior our algorithm enables efficient empirical tests of the robustness of the inference results. Since scientists cannot access the ground truth, these tests are necessary for trusting inference in real-world applications. We perform experiments on a marginalized version of the simulation-based inference benchmark and two complex and narrow posteriors, highlighting the simulator efficiency of our algorithm as well as the quality of the estimated marginal posteriors. Implementation on GitHub. [1]

## 1 Introduction

Parametric stochastic simulators are ubiquitous in science [1, 2, 3] and using them to solve the Bayesian inverse problem is of general interest. Likelihood-based methods like Markov chain Monte Carlo (MCMC) [4, 5] or nested sampling [6] are applicable when the likelihood is tractable. It is equally common that the likelihood is only implicitly defined by the simulator or is inefficient to compute. For this so-called *likelihood-free* or *simulation-based* inference, the traditional approach is Approximate Bayesian Computation (ABC) [7, 8]. See [9] for a reference.

Simulation-based inference (SBI) is closely connected to ABC and has been an open research topic since as early as the 1980s [10]. Deep learning has accelerated progress in the field [11, 12, 13, 14]. Proposed algorithms that learn the likelihood [13] or the posterior [14, 15, 16] utilize a density estimator. The likelihood-to-evidence ratio [11] can be learned via a classification-based technique. Refs. [11] and [14] were brought into a unified framework by [17].

High-fidelity simulators often have many parameters and/or an intractable likelihood function, which can make inference notoriously difficult. Practitioners are usually faced with observational data and an expensive stochastic simulator, without access to the ground truth posterior. They want a testably accurate posterior estimate without extreme simulation expense. With existing methods, the

---

[1]Implementation of experiments at https://github.com/bkmi/tmnre/. Ready-to-use implementation of underlying algorithm at https://github.com/undark-lab/swyft/.

Table 1: Comparison of SBI methods, including our proposed TMNRE, along with select properties. Properties listed are intended to showcase TMNRE and do not necessarily reflect the most desirable properties in every inference setting. For example, if cost were not a inhibiting factor a tractable joint distribution may be more appealing than targeting marginals directly. Similarly, a fully amortized posterior estimate is more flexible than a targeted one but remains, often, prohibitively expensive.

| Property / Method | Likelihood-based | ABC | NRE | NPE | SNRE | SNPE | **TMNRE** |
|---|---|---|---|---|---|---|---|
| Targeted inference | ✓ | • | ✗ | ✗ | ✓ | ✓ | ✓ |
| Simulator efficient *direct* marginals | ✗ | ✓ | • | • | ✗ | ✗ | ✓ |
| (Local) amortization | ✗ | ✗ | ✓ | ✓ | ✗ | ✗ | ✓ |

practitioner must choose between increased accuracy per simulation (so-called sequential methods [13, 18]) or efficient empirical testability (so-called amortized methods [11]). We provide a method which offers both simultaneously with a balance that can be tuned by a hyperparameter. Three attributes contribute to this goal:

***Targeted inference.*** Focusing simulations on the parameter regions that are most relevant for the inference problem and target data is more efficient. This is particularly true when most posterior density is concentrated compared to the prior's density.

***Marginal posteriors instead of the joint.*** Scientific insight is often based on a low dimensional marginalization of the posterior with nuisance parameters removed [19]. The additional information of the full joint posterior might not be worth the additional cost afforded. Targeting marginals directly, by estimating only the marginal for the parameters of interest, is simpler and sufficient for many scientific, parameter estimation, and bounding purposes.

***Consistency checks through local amortization.*** Practitioners are interested in testing the quality of inference methods [20, 21, 22]. One such test is to compare the empirical and nominal contained mass of estimated credible regions [23]. *Amortized* methods learn the posterior for any data, generated by any parameter, facilitating empirical study of the nominal credible regions on fabricated data. Still, learning an amortized posterior is excessive if only a small subset of parameters are consistent with a target observation.

We propose the concept of *local amortization* to learn the posterior on said subset, combining simulator efficiency of targeted inference with the testability of amortization. Both are critical components for enabling trustworthy scientific results.

**Our contribution.** We propose an algorithm that simultaneously achieves all three of the above aspects: Truncated Marginal Neural Ratio Estimation (TMNRE). It approximates the marginal likelihood-to-evidence ratio in a sequence of rounds and shines when the joint posterior is prohibitively costly. As a basis, we adopt likelihood-to-evidence ratio estimation proposed in [11], although our truncation scheme is applicable to other neural simulation-based inference methods which estimate the posterior or likelihood [13, 17]. Our iterative scheme is loosely inspired by likelihood-based nested sampling [6, 24, 25] since we generate training data drawn from a nested sequence of truncated priors in multiple rounds. Our algorithm (a) preferentially generates simulations in relevant regions of the parameter space, (b) allows estimation of all marginals of interest simultaneously and in parallel from the same training data, and (c) yields posteriors that are locally amortized in a constrained region around the posterior, enabling empirical self-consistency test of the inference results.

**Related work.** In Table 1, we compare the properties and features of a selection of deep-learning based simulation-based inference methods that are directly relevant for our work. Sampling from regions of highest probability density is baked into most *likelihood-based* methods [4, 5, 6, 24, 25, 26, 27, 28]. Amortization is generally not available with these methods because they sample from a particular posterior. *Approximate Bayesian Computation (*ABC*)* is a rejection sampling technique where proposed samples from the generative model are accepted based on a user defined distance criterion comparing generated data to the observation of interest. Two important methods include REJ-ABC [9] where the proposal distribution is simply the parameter prior and SMC-ABC [29, 30] where the proposal is iteratively refined. Blum and François [31] introduce an ABC distance criterion weighting mechanism to tune the posterior sampler as well as a proposal prior which draws from a truncated region of true prior. It estimates the support of previously-accepted samples via support vector machines [32] and samples from this region with rejection.

Likelihood-free inference can be cast as a conditional density estimation problem targeting either the posterior directly [14, 15, 16, 17] or the likelihood [13, 33]. This technique requires a density estimator, normally implemented as a mixture density network [34] or a normalizing flow [35, 36]. *Neural Likelihood Estimation* (NLE) performs well on benchmark tasks but must learn a density representation of the data in an unsupervised setting–difficult for complex data. Modern variants of *Neural Posterior Estimation* (NPE) [17] have become effective enough to offer an alternative estimation method for scientific practitioners [37].

*Amortized Approximate Ratio Estimators / Neural Ratio Estimation (*NRE*):* Binary classification allows estimation of the likelihood ratio between two hypotheses [12, 38, 39, 40, 41, 42] and was most famously applied to Generative Adversarial Networks [43]. Ref. [11] noted that naive application in the likelihood-free setting was unsatisfactory because the mathematically arbitrary choice of reference hypothesis significantly affected empirical MCMC results. Comparing likelihoods from jointly drawn $(\boldsymbol{x}, \boldsymbol{\theta}) \sim p(\boldsymbol{x}, \boldsymbol{\theta})$ and marginally drawn $(\boldsymbol{x}, \boldsymbol{\theta}) \sim p(\boldsymbol{x})p(\boldsymbol{\theta})$ samples, where $\boldsymbol{x}$ and $\boldsymbol{\theta}$ refer respectively to simulated data and simulator parameters, addresses the issue. Ref. [17] cast NRE and NPE in a unifying framework by adapting the loss function to contrast several possible hypothetical parameters. In this paper we refer to the algorithm described in [11] as NRE or NRE_A while the likelihood ratio algorithm described in [17] is referred to as NRE_B.

*Directly estimating the marginal posterior* distribution has been mentioned [11] and explored [44, 45]. Moment networks [45] produce the (central) moments of the posterior distribution, without calculating the density explicitly, via a hierarchy of neural networks trained on a regression problem. Ref. [45] also introduced a method which learns marginal posteriors with normalizing flows but does not address targeted inference or testability of estimated posteriors.

*Sequential Methods:* The neural likelihood-free methods generally offer a so-called sequential formulation that targets the posterior of a particular observation $\boldsymbol{x}_o$ [11, 13, 17]. Rather than drawing samples from the prior, the simulation budget is divided between rounds and the previous round's posterior is used as the new proposal distribution for the next round. This method increases simulation efficiency, but does not allow for amortization. Importantly, sequential methods can become highly inefficient when targeting multiple marginal posteriors directly because the previous round's marginal posterior does not update beliefs about the other parameters. A full parameter vector is necessary to run the simulator, thus defeating the purpose for all nuisance (marginalized-over) parameters.

## 2  Method

We aim to estimate any marginal posterior of interest using an approximate marginal likelihood-to-evidence ratio. Although we normally compute every one- and two-dimensional marginal posterior for visualization purposes, our method is not limited to this restriction. We define the object of study...

Let parametric stochastic simulator $\boldsymbol{g}$ be a nonlinear function that maps a vector of real parameters $\boldsymbol{\theta} = (\theta_1, \ldots, \theta_D)$ and a stochastic latent state $\boldsymbol{z} \in \mathbb{R}^{N_z}, N_z \in \mathbb{N}$ to an observation $\boldsymbol{x} = \boldsymbol{g}(\boldsymbol{\theta}, \boldsymbol{z})$. We consider a *factorizable prior* $p(\boldsymbol{\theta}) = p(\theta_1) \cdots p(\theta_D)$ over the parameters. The joint posterior is given via Bayes' rule as $p(\boldsymbol{\theta} \mid \boldsymbol{x}) = p(\boldsymbol{x} \mid \boldsymbol{\theta})p(\boldsymbol{\theta})/p(\boldsymbol{x})$, where $p(\boldsymbol{x} \mid \boldsymbol{\theta})$ is the intractable likelihood (or implicit distribution [10, 46]) and $p(\boldsymbol{x})$ is the evidence. Our goal is to efficiently compute arbitrary marginal posteriors, $p(\boldsymbol{\vartheta} \mid \boldsymbol{x})$. Here, $\boldsymbol{\vartheta}$ are the parameters of interest, and we denote all other (nuisance) parameters by $\boldsymbol{\eta}$, such that $\boldsymbol{\theta} = (\boldsymbol{\vartheta}, \boldsymbol{\eta})$. The marginal posterior is obtained from the joint distribution $p(\boldsymbol{\vartheta}, \boldsymbol{\eta} \mid \boldsymbol{x}) \coloneqq p(\boldsymbol{\theta} \mid \boldsymbol{x})$ by integrating over all components of $\boldsymbol{\eta}$,

$$p(\boldsymbol{\vartheta} \mid \boldsymbol{x}) = \int p(\boldsymbol{\vartheta}, \boldsymbol{\eta} \mid \boldsymbol{x}) \, d\boldsymbol{\eta} = \frac{\int p(\boldsymbol{x} \mid \boldsymbol{\vartheta}, \boldsymbol{\eta})p(\boldsymbol{\eta}) \, d\boldsymbol{\eta}}{p(\boldsymbol{x})} p(\boldsymbol{\vartheta}) \coloneqq \frac{p(\boldsymbol{x} \mid \boldsymbol{\vartheta})}{p(\boldsymbol{x})} p(\boldsymbol{\vartheta}). \tag{1}$$

where we used Bayes' rule, prior factorizibility, and defined the marginal likelihood $p(\boldsymbol{x} \mid \boldsymbol{\vartheta})$.

### 2.1  Marginal Neural Ratio Estimation (MNRE)

This paper considers the set of one- and two-dimensional marginal posteriors and their corresponding parameters of interest. Given parameter vector $\boldsymbol{\theta} \in \mathbb{R}^D$, define the set of all parameters associated with the one dimensional marginal posteriors by $\Theta_1 \coloneqq \{\theta_1, \ldots, \theta_D\}$. We do something similar, up to symmetry, for all two dimensional marginal posteriors $\Theta_2 \coloneqq \left\{ (\theta_i, \theta_j) \in \mathbb{R}^2 \mid i = 1, \ldots, D, \, j = i+1, \ldots, D \right\}$. We set our marginals of interest $\{\boldsymbol{\vartheta}_k\} \coloneqq$

$\Theta_1 \cup \Theta_2$ but in the general case, $\{\boldsymbol{\vartheta}_k\}$ can be any set of marginals that the practitioner desires. For every $\boldsymbol{\vartheta}_k$ we use NRE [11] to estimate the corresponding marginal likelihood-to-evidence ratio

$$r_k(\boldsymbol{x} \mid \boldsymbol{\vartheta}_k) := \frac{p(\boldsymbol{x} \mid \boldsymbol{\vartheta}_k)}{p(\boldsymbol{x})} = \frac{p(\boldsymbol{x}, \boldsymbol{\vartheta}_k)}{p(\boldsymbol{x})p(\boldsymbol{\vartheta}_k)} = \frac{p(\boldsymbol{\vartheta}_k \mid \boldsymbol{x})}{p(\boldsymbol{\vartheta}_k)} \ . \tag{2}$$

To this end, we train binary classifiers $\hat{\rho}_{k,\phi}(\boldsymbol{x}, \boldsymbol{\vartheta}_k)$ to distinguish jointly drawn parameter-simulation pairs $(\boldsymbol{x}, \boldsymbol{\vartheta}_k) \sim p(\boldsymbol{x}, \boldsymbol{\vartheta}_k)$ from marginally drawn parameter-simulation pairs $(\boldsymbol{x}, \boldsymbol{\vartheta}_k) \sim p(\boldsymbol{x})p(\boldsymbol{\vartheta}_k)$, where $\phi$ represents the parameters of the classifier. A Bayes optimal classifier $\rho_k$ would recover the density $\rho_k(\boldsymbol{x}, \boldsymbol{\vartheta}_k) = \frac{p(\boldsymbol{x}, \boldsymbol{\vartheta}_k)}{p(\boldsymbol{x}, \boldsymbol{\vartheta}_k) + p(\boldsymbol{x})p(\boldsymbol{\vartheta}_k)}$. Then the ratios of interest can be estimated by

$$\hat{r}_k(\boldsymbol{x} \mid \boldsymbol{\vartheta}_k) := \frac{\hat{\rho}_{k,\phi}(\boldsymbol{x}, \boldsymbol{\vartheta}_k)}{1 - \hat{\rho}_{k,\phi}(\boldsymbol{x}, \boldsymbol{\vartheta}_k)} \approx \frac{p(\boldsymbol{x}, \boldsymbol{\vartheta}_k)}{p(\boldsymbol{x})p(\boldsymbol{\vartheta}_k)} = r_k(\boldsymbol{x} \mid \boldsymbol{\vartheta}_k) \ . \tag{3}$$

We train each ratio estimator $\hat{r}_k(\boldsymbol{x} \mid \boldsymbol{\vartheta})$ using Adam [47] to minimize the binary cross-entropy (BCE)

$$\ell_k = - \int \left[ p(\boldsymbol{x} \mid \boldsymbol{\theta})p(\boldsymbol{\theta}) \ln \hat{\rho}_{k,\phi}(\boldsymbol{x}, \boldsymbol{\vartheta}_k) + p(\boldsymbol{x})p(\boldsymbol{\theta}) \ln \left( 1 - \hat{\rho}_{k,\phi}(\boldsymbol{x}, \boldsymbol{\vartheta}_k) \right) \right] d\boldsymbol{x} \, d\boldsymbol{\theta} \ . \tag{4}$$

In practice, we concatenate $\boldsymbol{x}$ with $\boldsymbol{\vartheta}_k$ as the input to $\hat{\rho}_{k,\phi}$. Since each classifier trains independently, it is trivial to train them all in parallel using the same underlying $(\boldsymbol{x}, \boldsymbol{\theta})$ pairs.

Practically, we parameterize the classifier by $\hat{\rho}_{k,\phi}(\boldsymbol{x}, \boldsymbol{\vartheta}_k) = \sigma \circ f_{k,\phi}(\boldsymbol{x}, \boldsymbol{\vartheta}_k)$, where $\sigma$ is the logistic sigmoid and $f_{k,\phi}$ is a neural network. The connection in Eq. (3) between the estimated ratio and the classifier implies that $\log \hat{r}_k(\boldsymbol{x} \mid \boldsymbol{\vartheta}_k) = f_{k,\phi}(\boldsymbol{x}, \boldsymbol{\vartheta}_k)$. We call the above technique MNRE.

When training data is limited, we found empirically (see Sec. 3.2 below) that the MNRE approach typically leads to conservative (i.e., not overconfident / not narrower than the ground truth) likelihood-to-evidence ratio estimates, provided early stopping criteria are used to avoid over-fitting of the classifier. At its core MNRE solves a simple, supervised binary classification task rather than a complex, unsupervised density estimation problem. Classification tasks are generally easier to train [12], and can rely on battle-tested network architectures.

## 2.2 Truncated Marginal Neural Ratio Estimation (TMNRE)

MNRE and NRE estimate a (marginal) likelihood-to-evidence ratio agnostic to the observed data $\boldsymbol{x}$ or parameter $\boldsymbol{\theta}$, a so-called *amortized* estimate. In other words, MNRE is suitable when $\boldsymbol{x} \in \left\{ \boldsymbol{g}(\boldsymbol{\theta}, \boldsymbol{z}) \mid \boldsymbol{\theta} \in \Omega, \boldsymbol{z} \in \mathbb{R}^{N_z} \right\}$ where $\Omega$ is the support of the prior. We propose an extension of this algorithm that enables targeted simulation of parameters relevant to a given target observation $\boldsymbol{x}_o$, and locally amortizes posteriors such that it enables empirical tests of the inference results. *Local amortization* implies that our proposed method is suitable when $\boldsymbol{x} \in \left\{ \boldsymbol{g}(\boldsymbol{\theta}, \boldsymbol{z}) \mid \boldsymbol{\theta} \in \Gamma, \boldsymbol{z} \in \mathbb{R}^{N_z} \right\}$ where the parameter region $\Gamma \subset \Omega$ is a function of $\boldsymbol{x}_o$ and will be defined below.

We observe that values of $\boldsymbol{\theta}$ which could not have plausibly generated $\boldsymbol{x}_o$ evaluate to negligible posterior density, i.e. $p(\boldsymbol{\theta} \mid \boldsymbol{x}_o) \approx 0$, which suggests that the corresponding parameters $\boldsymbol{\theta}$ do not significantly contribute to the marginalization in Eq. (1). We denote a prior that is suitably constrained to parameters with non-negligible posterior density $p(\boldsymbol{\theta} \mid \boldsymbol{x}_o)$ by

$$p_\Gamma(\boldsymbol{\theta}) := V^{-1} \mathbb{1}_\Gamma(\boldsymbol{\theta}) p(\boldsymbol{\theta}) \ , \tag{5}$$

where $\mathbb{1}_\Gamma(\boldsymbol{\theta})$ is an indicator function that is unity on $\Gamma \subset \Omega$ and zero otherwise, and $V^{-1}$ is a normalizing constant (which can be interpreted as the fractional volume of the truncated prior). The subscript $\square_\Gamma$ denotes that arbitrary symbol $\square$ is based on a prior truncated by indicator function $\mathbb{1}_\Gamma$.

We define a rectangular indicator function $\mathbb{1}_{\Gamma^{\text{rec}}}$ by discarding parameters that lie in the far tails of the one dimensional marginal posteriors of our target observation $\boldsymbol{x}_o$, using a thresholding $\epsilon \ll 1$, via

$$\Gamma^{\text{rec}} = \left\{ \boldsymbol{\theta} \in \Omega \ \middle| \ \forall d = 1, \ldots, D : \frac{p(\theta_d \mid \boldsymbol{x}_o)}{\max_{\theta_d} p(\theta_d \mid \boldsymbol{x}_o)} > \epsilon \right\} \ . \tag{6}$$

For Gaussian joint posteriors, this scheme leads to one dimensional marginal posteriors $p_\Gamma(\theta_d \mid \boldsymbol{x}_o)$ that are truncated at their approximately $\pm\sqrt{-2\ln\epsilon}\,\sigma$ tail. In general, truncation will lead to an approximation error that can be estimated as $p_{\Gamma^{\text{rec}}}(\boldsymbol{\theta} \mid \boldsymbol{x}_o) = p(\boldsymbol{\theta} \mid \boldsymbol{x}_o) + \mathcal{O}(\epsilon) \max_{\boldsymbol{\theta}} p(\boldsymbol{\theta} \mid \boldsymbol{x}_o)$, see Appendix C. Throughout this paper we use $\epsilon = 10^{-6}$, which corresponds to $\pm 5.26\sigma$ for a Gaussian

---

**Algorithm 1** Truncated Marginal Neural Ratio Estimation (TMNRE)

---

*Inputs:*      Simulator $p(\boldsymbol{x} \mid \boldsymbol{\theta})$, factorizable prior $p(\boldsymbol{\theta})$, real observation $\boldsymbol{x}_0$, max rounds $M$, training data per round $N^{(m)}$, threshold $\epsilon$, dimension of parameters $D$, mass ratio $\beta$, classifiers $\boldsymbol{\rho}_1(\boldsymbol{x}, \boldsymbol{\theta}) = \{\sigma \circ f_{\phi,d}(\boldsymbol{x}, \theta_d)\}_{d=1}^{D}$ and $\boldsymbol{\rho}_2(\boldsymbol{x}, \boldsymbol{\theta}) = \{\sigma \circ f_{\phi,d}(\boldsymbol{x}, \vartheta_d)\}_{d=(1,1)}^{(D,D)}$.

*Outputs:*    Parameterized classifiers $\boldsymbol{\rho}_1(\boldsymbol{x}, \boldsymbol{\theta})$ and $\boldsymbol{\rho}_2(\boldsymbol{x}, \boldsymbol{\theta})$, constrained region $\Gamma^{\mathrm{rec}}$.

---

1: **procedure** MNRE($\mathcal{D}, \boldsymbol{\theta}', \boldsymbol{\rho}_\phi$)
2:    **while** $\boldsymbol{\rho}_\phi$ not converged **do**
3:      $\phi \leftarrow \mathrm{OPTIMIZER}\left(\phi, \nabla_\phi \sum_k \left[\mathrm{BCE}(\hat{\rho}_{\phi,k}(\boldsymbol{x}, \vartheta_k), 1) + \mathrm{BCE}(\hat{\rho}_{\phi,k}(\boldsymbol{x}, \vartheta'_k), 0)\right]\right)$
4:    **return** $\mathbf{f}_\phi$

*Initialize:*    $\mathcal{D}^{(0)} \leftarrow \{\}$,   $\Gamma^{(0)} \leftarrow \mathrm{supp}(p(\boldsymbol{\theta}))$,   $\alpha^{(0)} \leftarrow 0$,   $m \leftarrow 1$.

1: **procedure** TMNRE
2:    **while** $\alpha^{(m-1)} \leq \beta$ and $m \leq M$ **do**
3:      $\mathcal{D}_\Gamma^{(m-1)} \leftarrow \left\{(\boldsymbol{x}^{(n)}, \boldsymbol{\theta}^{(n)}) \in \mathcal{D}^{(m-1)} \mid \boldsymbol{\theta}^{(n)} \in \Gamma^{(m-1)}\right\}$      ▷ *Retain data in region*
4:      $N_{\mathrm{simulate}}^{(m)} \leftarrow N^{(m)} - |\mathcal{D}_\Gamma^{(m-1)}|$      ▷ *Calculate num. necessary simulations*
5:      $\boldsymbol{\theta} \leftarrow \{\boldsymbol{\theta}^{(n)} \sim \mathbb{1}_{\Gamma^{(m-1)}}(\boldsymbol{\theta}) p(\boldsymbol{\theta})\}_{n=1}^{N_{\mathrm{simulate}}^{(m)}}$      ▷ *Sample for jointly distributed pairs*
6:      $\boldsymbol{x} \leftarrow \{\boldsymbol{x}^{(n)} \sim p(\boldsymbol{x} \mid \boldsymbol{\theta}^{(n)})\}_{n=1}^{N_{\mathrm{simulate}}^{(m)}}$      ▷ *Simulate jointly distributed pairs*
7:      $\boldsymbol{\theta}' \leftarrow \{\boldsymbol{\theta}^{(n)} \sim \mathbb{1}_{\Gamma^{(m-1)}}(\boldsymbol{\theta}) p(\boldsymbol{\theta})\}_{n=1}^{N^{(m)}}$      ▷ *Sample for marginally distributed pairs*
8:      $\mathcal{D}^{(m)} \leftarrow \mathcal{D}_\Gamma^{(m-1)} \cup \{(\boldsymbol{x}^{(n)}, \boldsymbol{\theta}^{(n)})\}_{n=1}^{N_{\mathrm{simulate}}^{(m)}}$      ▷ *Aggregate training data*
9:      $\boldsymbol{\rho}_1 \leftarrow \mathrm{MNRE}(\mathcal{D}^{(m)}, \boldsymbol{\theta}', \boldsymbol{\rho}_1)$
10:      $\Gamma^{(m)} \leftarrow \left\{\boldsymbol{\theta} \in \Gamma^{(m-1)} \mid \forall d : \frac{\hat{p}_{d,\Gamma^{(m)}}(\theta_d | \boldsymbol{x}_o)}{\max_{\theta_d} \hat{p}_{d,\Gamma^{(m)}}(\theta_d | \boldsymbol{x}_o)} > \epsilon\right\}$      ▷ *Find constrained region*
11:      $\alpha^{(m)} \leftarrow \int \mathbb{1}_{\Gamma^{(m)}}(\boldsymbol{\theta}) p(\boldsymbol{\theta}) d\boldsymbol{\theta} / \int \mathbb{1}_{\Gamma^{(m-1)}}(\boldsymbol{\theta}) p(\boldsymbol{\theta}) d\boldsymbol{\theta}$      ▷ *Update prior mass ratio*
12:      $m \leftarrow m + 1$      ▷ *Increment counter*
13:    $\boldsymbol{\rho}_2 \leftarrow \mathrm{MNRE}(\mathcal{D}^{(m)}, \boldsymbol{\theta}', \boldsymbol{\rho}_2)$
14:    **return** $\boldsymbol{\rho}_1, \boldsymbol{\rho}_2, \Gamma^{(m)}$

---

posterior. Those truncations do not affect the location of high-probability credible contours and have hence no practical effect on parameter inference tasks. We provide more exemplary error estimates for a range of cases in Appendix C.

Our algorithm defines a series of nested indicator functions $\mathbb{1}_{\Gamma^{(m)}}$ whose regions have the property

$$\Omega := \Gamma^{(1)} \supset \Gamma^{(2)} \supset \cdots \supset \Gamma^{(M)} \supset \Gamma^{\mathrm{rec}}. \tag{7}$$

They iteratively approximate the indicator function $\mathbb{1}_{\Gamma^{\mathrm{rec}}}$ in multiple rounds $m = 1, \ldots, M$. This sequence is generated with the following steps:

- We initialize $\Gamma^{(1)} = \Omega := \mathrm{supp}(p(\boldsymbol{\theta}))$, meaning that we start with the unconstrained prior.

- Each round $1 \leq m \leq M$, we train $D$, one dimensional ratio estimators $\hat{r}_{d,\Gamma^{(m)}}(\boldsymbol{x} \mid \theta_d)$ using data from within the constrained region, $\boldsymbol{\theta} \in \Gamma^{(m)}$. The estimated marginal posterior is $\hat{p}_{\Gamma^{(m)}}(\theta_d \mid \boldsymbol{x}) = \hat{r}_{d,\Gamma^{(m)}}(\boldsymbol{x} \mid \theta_d) p_{\Gamma^{(m)}}(\theta_d)$. To this end, do MNRE, setting $\vartheta_k = \theta_k$, $d \in \{1, 2, \ldots, D\}$ using the constrained prior $p_{\Gamma^{(m)}}(\boldsymbol{\theta})$ with $N^{(m)}$ training samples per round.

- For each round $m < M$, we estimate the indicator function for the next round using the approximated posteriors, via

$$\Gamma^{(m+1)} = \left\{\boldsymbol{\theta} \in \Gamma^{(m)} \mid \forall d : \frac{\hat{p}_{\Gamma^{(m)}}(\theta_d \mid \boldsymbol{x}_o)}{\max_{\theta_d} \hat{p}_{\Gamma^{(m)}}(\theta_d \mid \boldsymbol{x}_o)} > \epsilon\right\} . \tag{8}$$

- The last round is determined either when $m = M$ or when a stopping criterion is reached. The stopping criterion is defined by the ratio of consecutive truncated prior masses. It is satisfied when the sequence of truncated priors have the property $\int \mathbb{1}_{\Gamma^{(m)}}(\boldsymbol{\theta}) p(\boldsymbol{\theta}) d\boldsymbol{\theta} / \int \mathbb{1}_{\Gamma^{(m-1)}}(\boldsymbol{\theta}) p(\boldsymbol{\theta}) d\boldsymbol{\theta} > \beta$. We often set $\beta = 0.8$.

- Using the data from this final constrained region, we can approximate any marginal posterior of interest. In this paper, we estimate the one- and two-dimensional marginals necessary for corner plots. We emphasize that the data already generated during the truncation phase can be reused to learn arbitrary marginals of interest. When higher accuracy likelihood-to-evidence ratio estimates are needed, the user can simulate from the truncated region.

We briefly address failure modes. First, this algorithm relies on the assumption that posterior estimates $\hat{p}_{\Gamma^{(m)}}(\theta_d \mid \boldsymbol{x}_o)$ from MNRE provide a good approximation of $p(\theta_d \mid \boldsymbol{x}_o)$. An over-confident estimate would remove parameter ranges that are part of $\Gamma^{\text{rec}}$. In practice, we have not observed this effect, although it is a concern with all simulation-based methods [23]. We give credit to early stopping and a conservative choice of $\epsilon$, and provide further illustration and support in Sec. 3 below. Second, since the truncated posterior only agrees with the ground truth up to corrections of order $\epsilon$, the iterative scheme will not converge to Eq. (6); rather to a similar expression where the right-hand side of the inequality in Eq. (6) receives additional $\mathcal{O}(\epsilon)$ corrections. Although these corrections mildly affect the truncations, they are of little practical relevance since we choose an $\epsilon$ which is very small. Both failure modes are diagnosed by checking whether high probability regions of the estimated posteriors intersect with the boundaries of the indicator function.

Algorithm 1 estimates the necessary marginal posteriors for corner plot visualization. We demonstrate the cost-effectiveness of this algorithm in Section 3. An important limitation of Algorithm 1 is the inaccessibility of the posterior predictive distribution. This limitation is mitigated by training a ratio estimator on all parameters within the truncated region; however, producing an accurate joint estimate may come with (sometimes significant) additional simulation costs.

Like sequential methods [13, 17] the number of rounds $M$, the training data per round $N^{(m)}$, and any stopping criteria $\beta$ are hyperparameters. For further discussion and default values see Appendix A, for bound derivations and limitations see Appendix C and D. We present TMNRE in Algorithm 1.

**Properties of our algorithm.** We discuss the properties of our algorithm in support of Table 1. First, our algorithm performs *targeted inference* by successively focusing on regions of the parameter space that are compatible with an observation $\boldsymbol{x}_o$. Second, since training data is always drawn from the prior, it is possible to efficiently *train arbitrary marginal posteriors* with the same training data generated for round $M$. Third, the algorithm trains *locally amortized posteriors* that are valid for parameters $\boldsymbol{\theta} \in \Gamma^{(m)}$, facilitating empirical consistency checks of the estimated posteriors within this region. TMNRE's properties provide a favorable cost-benefit ratio–yielding marginal insight into the posterior without paying the price of accessing the joint posterior. The price of the joint is often inhibiting for expensive simulators. These aspects will be demonstrated in the experiments in the following section.

## 3 Experiments

First, we perform experiments to compare TMNRE to other algorithms on standard benchmarks from the simulation-based inference literature. Next, we highlight useful aspects of our algorithm regarding targeted inference, marginalization and local amortization with two additional experiments. These experiments compare algorithms using performance metrics which access the ground truth posterior. For practitioners who normally cannot access the ground truth, TMNRE offers an empirical consistency check (see Section 3.3) to assess the quality of the estimated posterior. Such a check is impractical for sequential methods and is one of the primary practical applications of TMNRE. Further experiments, including application on a cosmology simulator, can be found in Appendix E.

### 3.1 Performance on standard tasks

We compare the performance of our algorithm with other traditional and neural simulation-based inference methods on a selection of problems from the SBI benchmark [18]. Each *task* is defined by a simulator, ten observations, a simulation budget, and 10,000 samples drawn from corresponding reference and approximate posterior distributions. The reference samples enable quantification of algorithmic accuracy on a range of performance metrics. We evaluate performance on all tasks except the Bernoulli Generalized Linear Model because its prior is not factorizable. Details in Appendix A.

Since our method estimates every one- and two-dimensional marginal posterior, we compare samples from our approximate marginal posteriors with samples from the reference joint posterior, marginalized over nuisance parameters. We quantify the results using the Classifier 2-Sample Test [48, 49]. We train a C2ST classifier for each of the $\binom{D}{1} + \binom{D}{2}$ possible one- and two-dimensional marginals, and report the mean values, and 95% confidence intervals, in Figure 1. We call this averaged performance metric *C2ST-ddm*, see Appendix B for more detail. The results are presented as grouped by dimensionality since difficulty increases with dimension and is reflected in the C2ST-ddm scores.

For comparison, we computed the C2ST-ddm on the other benchmark methods' posterior samples. Unlike our method, which was trained on the marginals directly, the benchmark methods were trained to estimate the joint posterior. We note that since TM-NRE trains a neural network for every marginal (efficiently, in parallel), our method has many times more parameters than any neural likelihood-free inference method that directly targets the joint. However, parameter count is not a scarce resource in SBI. Training hyperparameters can be found in Appendix A.

The maximum number of rounds before meeting the stopping criteria varied across tasks from just one round with no truncation up to seven rounds on Gaussian Linear. Out of 240 runs, five did not converge for TMNRE.

As shown in Figure 1, our method outperformed REJ-ABC and SMC-ABC and offers increased efficiency compared to non-sequential methods on all tasks, except Gaussian Linear. On some tasks, TMNRE is competitive with sequential methods. Generally, TMNRE performs best on narrow posteriors and a large simulation budget. Benefits diminished on tasks with wide posteriors like Gaussian Linear, SLCP, and SLCP Distractors–a limitation of TMNRE. Based on these results, ours is the only method, among sequential and non-sequential techniques, which offers both sufficient accuracy and local amortization.

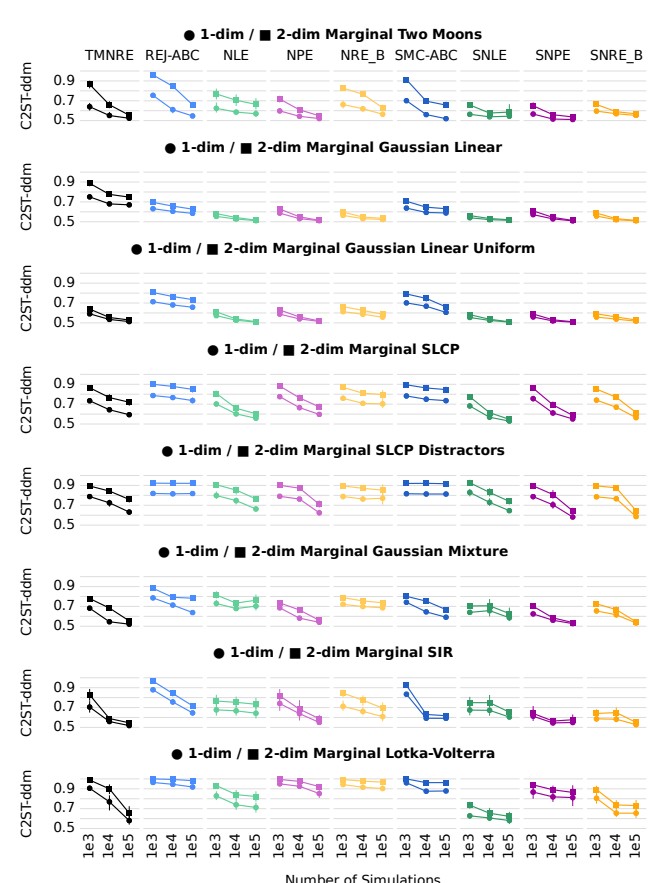

Figure 1: **Performance on marginalized posterior benchmark tasks.** Mean and 95% confidence intervals of classification accuracy (C2ST-ddm) for our method, TMNRE, and other SBI methods with 10 observations / budget. one- and two-dimensional scores are plotted. Lower scores imply better posteriors. Our simulation budget is approximate, see Appendix A. The plot and tasks are derivative of [18].

## 3.2 Efficient targeted inference: a 3-dim torus model

We define a task which highlights the effectiveness of truncating the prior, namely a simulator with a very small torus shaped posterior. We present an ablation study of the truncation method along with a hyperparameter scan of $\epsilon$. Task details and additional experiments are presented in Appendix A.

We ran Algorithm 1 which satisfied the stopping criterion after four rounds. We performed marginal likelihood-to-evidence ratio estimation on all one- and two-dimensional marginals for each step in the sequence of constrained regions, using the number of samples available that round. We also trained an estimator which used the same simulation budget but the samples were drawn from the unconstrained prior. We analyzed the prior volume, C2ST-ddm, and the sum of one dimensional KL divergences at

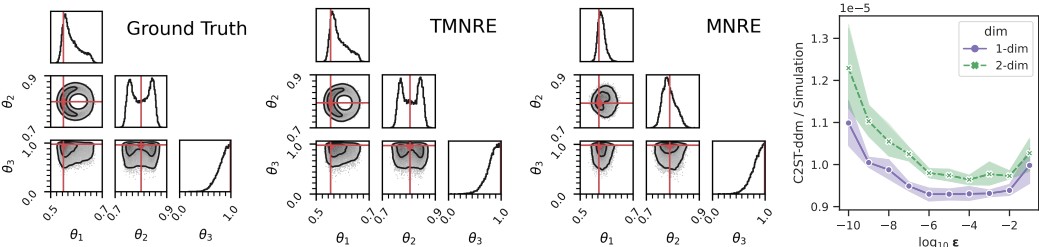

Figure 2: *First through Third Panel:* View of torus marginal posteriors as estimated by rejection sampling, TMNRE, and MNRE. In some dimensions, the posterior extends the full unit cube prior width, while in others it is very narrow. TMNRE easily finds the asymmetric details after constraining to the relevant region while MNRE does not. *Fourth Panel:* The results of a hyperparameter scan of $\epsilon$ on the torus task. The C2ST-ddm per simulation is reported versus $\epsilon$. The mean and 95% confidence intervals are shown over five repetitions of the experiment. Lower values indicate better performance.

each round for both methods. The posteriors are shown in Figure 2 and the performance metrics for the ablation study, are shown in Figure 4.

We found TMNRE very accurately approximated all marginals at the maximum simulation budget. MNRE placed mass in the correct region but missed the shape of the posterior entirely. TMNRE improved simulation efficiency compared with MNRE as indicated by the slope of the C2ST-ddm. The max-normalized posterior estimates at every round are plotted in Figure 3. We note that given the limited training data in early rounds, our method predicts wider posteriors than the ground truth. These are called conservative posterior estimates and they are the preferred failure mode for practitioners [23]. Other SBI methods are tested on the torus in Appendix A.

To determine the effects the hyperparameter $\epsilon$, we performed a grid search between $10^{-10}$ to $10^{-1}$ using TMNRE on the same simulator. We reported the performance in terms of the C2ST-ddm per simulation in Figure 2 and repeated the experiment five times. We observe that the optimal value of $\epsilon$ was $10^{-6}$ since it was the most conservative value of $\epsilon$ that optimized the metric.

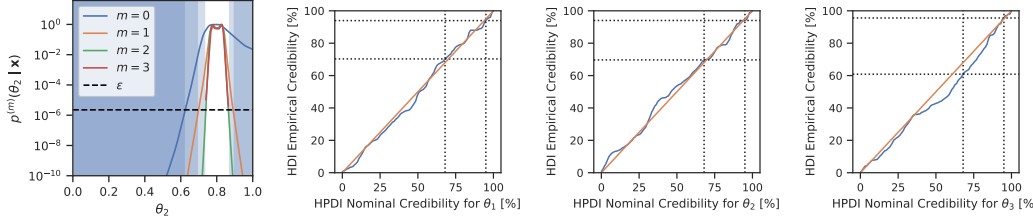

Figure 3: *First panel:* The ratio estimator for $\theta_2$ in the 3-dim torus example, for several consecutive rounds. The estimator is conservative with limited data because each round's estimate converges from above to the final results (red line); i.e. the early rounds produce too-wide posteriors thus truncation decisions are made safely. *Second to Fourth Panels:* Empirical versus nominal credibility for the highest posterior density intervals (HPDI) for each $\theta_d$ in the 3-dim torus. The line at (above) the diagonal implies accurate (conservative or wide) nominal credible intervals. See section 3.3.

### 3.3 Empirical tests of inference results through local amortization

Our algorithm locally amortizes the posterior for parameters drawn from the constrained prior $p_\Gamma(\boldsymbol{\theta})$. This opens the door for various experimental diagnostics to test the reliability of our trained inference networks with simulated data (which is also possible for NRE [50], but not for sequential methods that are exclusively targeting on one specific observation rather than a range of observations). We demonstrate this by comparing the empirical credibility to the nominal credibility for the highest posterior density intervals, similarly to [23] but we estimate using the truncated prior.

For the 3-dim torus example, we draw 10000 samples $(\boldsymbol{x}, \theta_d) \sim p(\boldsymbol{x} \mid \boldsymbol{\theta})p_\Gamma(\boldsymbol{\theta})$ from the truncated generative model. For all samples, we generate marginal posteriors, $\hat{p}(\theta_d \mid \boldsymbol{x})$. For those marginal

posteriors, we then derive the frequency with which $t\%$ highest density intervals contain the true value $\theta_d$. The result is shown in Fig. 3. It provides an immediate check of the reliability of our trained inference networks *without knowing the ground truth*, and provides a safeguard against overconfident statements, which is critical for using the results of inference networks in a scientific context.

This empirical test is designed to show the consistency of the estimated nominal credible intervals across realizations of fabricated data but it does not address a generative model mismatch or access whether the estimated posterior corresponds to the ground truth. When $\epsilon$ is small enough, the effects of truncation will not significantly impact this empirical test because of the accurate posterior estimate. Large $\epsilon$, that trim the tails of the estimated posterior aggressively, render this test unreliable because the estimated likelihood-to-evidence ratios will have inaccurate highest density intervals. It is possible to check whether $\epsilon$ is too large by observing a high-density posterior equicontour intersecting with a truncation bound, see Appendix D.

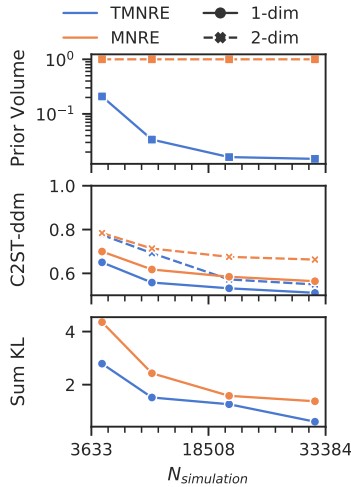

### 3.4 Efficient marginal posteriors: the 10-dim eggbox

We define a posterior that, when plotted in two dimensions, looks like a top-down view of an $2 \times 2$ eggbox. Let $\boldsymbol{\theta}, \boldsymbol{g}(\boldsymbol{\theta}) \in \mathbb{R}^D$ and $\theta_k$ denote the $k$th element of $\boldsymbol{\theta}$, then the simulator for this problem is defined $g_k(\boldsymbol{\theta}) = \sin(\theta_k \cdot \pi)$. To fix the posterior shape, we set $\theta_{k,o} = \frac{1}{4}$, $k = 1, 2, \ldots, D$ and $\boldsymbol{x}_0 = \boldsymbol{g}(\boldsymbol{\theta}_o)$. The likelihood is determined by an additive noise model $p(\boldsymbol{x} \mid \boldsymbol{\theta}) = \mathcal{N}(\boldsymbol{g}(\boldsymbol{\theta}), \sigma^2 \boldsymbol{I})$ with $\sigma = 0.1$. The total number of modes in our 10-dimensional model is $2^{10} = 1024$. Realistic models do not typically feature such a regular mode structure, but this pattern enables comparison of the various algorithm's ability to handle multimodal data.

Given 10,000 training samples drawn from the prior and a $D = 10$ dimensional parameter space, we trained MNRE to estimate all one- and two-dimensional marginals, the SBI [51]

Figure 4: Performance metrics on MNRE and TMNRE versus simulation budget. Budgets determined by truncation algorithm. *T:* Prior volume. *M:* According to C2ST-ddm, TMNRE produces more accurate posteriors. *L:* KL divergence summed over 1-dim marginals; same result.

implementation of NRE and SNRE on the joint, and finally a marginalized version of SNRE (SMNRE). In SMNRE, we divided the samples across 10 rounds and each round proposed samples according to the previous round's posterior distribution for the predicted marginals, but the initial prior for the nuisance parameters. Since, in a general setting, SMNRE cannot use samples from another marginal estimator, we divided the 10,000 training samples evenly among the 55, one- and two-dimensional marginal estimators, each estimator receiving 181 training samples. 25,000 samples from each reported posterior are visible in Figure 5. TMNRE recovered the structure of the ground truth marginal posteriors, providing empirical evidence that estimating marginals directly can provide high accuracy at low simulation budgets for complex high-dimensional posteriors. Experiments using other SBI methods are in Appendix A along with discussion of a rotated version of the problem.

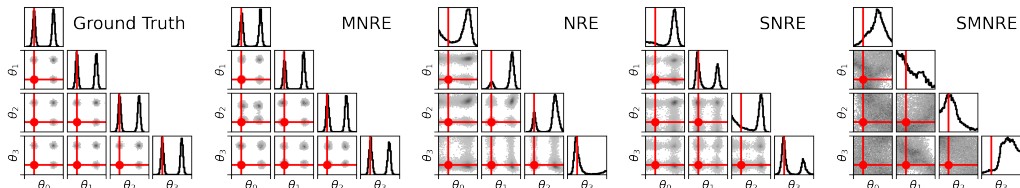

Figure 5: Posteriors from the 10-dim eggbox benchmark (only 4 parameters are shown for clarity). All methods received 10k training samples and produced 25k posterior samples. NRE and SNRE were trained jointly, while MNRE and SMNRE were trained marginally. SMNRE cannot share training samples between marginals so each estimator received an equal share of the total simulation budget, 181 samples. Our method recovered the structure of the ground truth, unlike NRE, SNRE, or SMNRE.

### 3.5 Practitioner's use case: cosmological inference with a simulator

Parameter inference is commonplace in astrophysics and cosmology. We are interested in a simulator with six parameters that returns three angular power spectra, as they would be measured by an idealized Cosmic Microwave Background (CMB) experiment. A budget of 5,000 simulations is sufficient for MNRE to accurately estimate the corner plot, while NRE produces unconstrained and inaccurate marginal posterior estimates. Due to space constraints, we could not include this practical application of MNRE in the body of this paper. Please see Appendix E for details of the simulator, the inference technique, and the results. A complete study of the topic is currently in preparation [52].

## 4 Discussion and conclusions

We presented Truncated Marginal Neural Ratio Estimation (TMNRE), a simulation-based inference algorithm based on NRE. The core idea of our algorithm is to focus on most probable parameter regions by truncating marginal posteriors in their very low-probability tails. For Gaussian posteriors this is typically beyond $5\sigma$ and does not significantly affect the higher density contours. In addition to performing on par or better than existing algorithms on standard benchmarks, TMNRE is better suited to the practitioner's needs than other algorithms because it offers simulation efficient marginal posterior estimation and the capacity to perform efficient consistency checks through local amortization. These features are particularly desirable to scientists whose simulators are expensive and rife with nuisance parameters.

TMNRE uses a sequence of training and sampling rounds to automatically produce parameters with high posterior density, i.e. relevant to a particular observation $x_o$. The output of this sequence is a hyperrectangular approximation to a highest posterior density region, implicitly defined by hyperparameter $\epsilon$. That implies parameters from within this constrained region are more likely to produce data similar to $x_o$ than simulations from outside the region. Using data drawn from this constrained region, TMNRE estimates any marginal posterior of interest directly using a marginal likelihood-to-evidence ratio; a simpler and more practical technique than estimating the entire joint posterior. Finally, by construction, our targeted inference method can accurately estimate posteriors of simulations from within the constrained parameter region. This freedom facilitates fast empirical studies of the nominal credible regions, which are of critical importance in real-life applications when there is no ground truth posterior to compare to.

On the SBI benchmark [18], we found that TMNRE is on par with the most effective SBI algorithms, such as SNRE [11], as measured by the C2ST performance metric, Fig. 1. We highlighted the benefits of TMNRE using two showcase tasks: a torus-shaped posterior and a multimodal eggbox-shaped posterior. The torus featured a very narrow posterior that TMNRE found and accurately learned while simple MNRE failed to do so, Fig. 4. We demonstrated validity of our iterative procedure, and the ability to perform important validation tests by testing the nominal credible intervals empirically, Fig. 3. The eggbox's joint posterior was challenging for NRE-based methods but MNRE efficiently estimated the marginal posteriors, Fig. 5. Further torus and eggbox experiments are in Appendix A and a cosmology posterior is estimated in Appendix E.

The presented algorithm is aimed at marginal posterior inference, which is a typical goal for scientific applications, but does not allow, e.g., to evaluate the posterior predictive distribution which requires the joint posterior. Furthermore, our algorithm particularly shines for high-dimensional problems with complex and/or narrow posteriors, whereas we expect that simpler problems may be better suited to other SBI methods. We address further limitations of our method in Appendix D.

We note that the hyperrectangular indicator function, defined in Eq. (8), is not optimal if some of the parameters are strongly correlated. However, it can be straightforwardly extended to more complex shapes. The challenge is to efficiently define the boundaries of the indicator function and sample from within it, a problem tackled by effective nested sampling algorithms [53].

This work is primarily foundational and the societal impacts, other than the cost of training machine learning models, would therefore be drawn from a hypothetical application. As this is an inference method, it would be possible to apply it to biased simulators which could reinforce unethical patterns. In general, the societal impacts are closely tied to the implications of the simulators themselves.

## Acknowledgments and Disclosure of Funding

This work uses `numpy` [54], `scipy` [55], `seaborn` [56], `matplotlib` [57], `altair` [58, 59], `pandas` [60, 61] `pytorch` [62], and `jupyter` [63]. Benjamin Kurt Miller is funded by the University of Amsterdam Faculty of Science (FNWI), Informatics Institute (IvI), and the Institute of Physics (IoP).

We want to thank the DAS-5 computing cluster for access to their TitanX GPUs. DAS-5 is funded by the NWO/NCF (the Netherlands Organization for Scientific Research). We received funding from the European Research Council (ERC) under the European Union's Horizon 2020 research and innovation programme (Grant agreement No. 864035 – UnDark).

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
