# A Experiments

In this section we present the relevant experimental details including a comprehensive list of experiments in Table 2. We first discuss the computational setting and approximate computational cost. Afterwards the details for the SBI benchmark, the torus, and the eggbox are presented along with additional experiments for those problems. Finally we discuss how we generate our datasets and how we can use the estimated likelihood-to-evidence ratio in practice. In all tests, we applied TMNRE with the hyperparameters laid out in Table 3.

Table 2: Experiments

| Task | Parameter Dimension | Algorithm | Simulations | Trained Marginalization | Evaluated Marginalization | Constrained | Metric |
|---|---|---|---|---|---|---|---|
| Two Moons | 2 | TMNRE | $\{1, 10, 100\}$E3 | 1d, 2d | 1d, 2d | 1d | C2ST-ddm |
| Two Moons | 2 | SBI | $\{1, 10, 100\}$E3 | Joint | 1d, 2d | | C2ST-ddm |
| Gaussian Linear | 10 | TMNRE | $\{1, 10, 100\}$E3 | 1d, 2d | 1d, 2d | 1d | C2ST-ddm |
| Gaussian Linear | 10 | SBI | $\{1, 10, 100\}$E3 | Joint | 1d, 2d | | C2ST-ddm |
| Gaussian Linear Uniform | 10 | TMNRE | $\{1, 10, 100\}$E3 | 1d, 2d | 1d, 2d | 1d | C2ST-ddm |
| Gaussian Linear Uniform | 10 | SBI | $\{1, 10, 100\}$E3 | Joint | 1d, 2d | | C2ST-ddm |
| SLCP | 5 | TMNRE | $\{1, 10, 100\}$E3 | 1d, 2d | 1d, 2d | 1d | C2ST-ddm |
| SLCP | 5 | SBI | $\{1, 10, 100\}$E3 | Joint | 1d, 2d | | C2ST-ddm |
| SLCP (distractors) | 5 | TMNRE | $\{1, 10, 100\}$E3 | 1d, 2d | 1d, 2d | 1d | C2ST-ddm |
| SLCP (distractors) | 5 | SBI | $\{1, 10, 100\}$E3 | Joint | 1d, 2d | | C2ST-ddm |
| Gaussian Mixture | 2 | TMNRE | $\{1, 10, 100\}$E3 | 1d, 2d | 1d, 2d | 1d | C2ST-ddm |
| Gaussian Mixture | 2 | SBI | $\{1, 10, 100\}$E3 | Joint | 1d, 2d | | C2ST-ddm |
| SIR | 2 | TMNRE | $\{1, 10, 100\}$E3 | 1d, 2d | 1d, 2d | 1d | C2ST-ddm |
| SIR | 2 | SBI | $\{1, 10, 100\}$E3 | Joint | 1d, 2d | | C2ST-ddm |
| Lotka-Volterra | 4 | TMNRE | $\{1, 10, 100\}$E3 | 1d, 2d | 1d, 2d | 1d | C2ST-ddm |
| Lotka-Volterra | 2 | SBI | $\{1, 10, 100\}$E3 | Joint | 1d, 2d | | C2ST-ddm |
| Torus | 3 | TMNRE | 4985, 11322, 21127, 32032 | 1d, 2d | 1d, 2d | 1d | C2ST-ddm, KLD, Visual |
| Torus | 3 | MNRE | 4985, 11322, 21127, 32032 | 1d, 2d | 1d, 2d | | C2ST-ddm, KLD, Visual |
| Torus | 3 | SBI | 4985, 11322, 21127, 32032 | Joint | | | Visual |
| Torus (epsilon scan) | 3 | TMNRE | $\sim$ 30 E3 | 1d, 2d | 1d, 2d | 1d | C2ST-ddm / simulation |
| Egg Box 2 modes / dim | 10 | MNRE | 10 E3 | 1d, 2d | | | Visual |
| Egg Box 2 modes / dim | 10 | NRE | 10 E3 | Joint | | | Visual |
| Egg Box 2 modes / dim | 10 | SNRE | 10 E3 | Joint | | | Visual |
| Egg Box 2 modes / dim | 10 | SMNRE | 10 E3 | 1d, 2d | | | Visual |
| Egg Box 2 modes / dim | 10 | Remaining SBI | 10 E3 | Joint | | | Visual |
| Rotated Egg Box | 10 | MNRE | 10 E3 | 1d, 2d | | | Visual |
| Rotated Egg Box | 10 | SBI | 10 E3 | Joint | | | Visual |

Table 3: TMNRE Hyperparameters

| Hyperparameter | Value |
|---|---|
| Activation Function | RELU |
| AMSGRAD | No |
| Architecture | RESNET (2 blocks) |
| Batch normalization | Yes |
| Batch size | 128 |
| Criterion | BCE |
| Dropout | No |
| Early stopping patience | 20 |
| $\epsilon$ | $e^{-13} \approx 10^{-6}$ |
| Hidden features | 64 |
| Percent validation | 10% |
| Reduce lr factor | 0.1 |
| Reduce lr patience | 5 |
| Max epochs | 300 |
| Max rounds | 10 |
| Learning rate | 0.01 |
| Learning rate scheduling | Decay on plateau |
| Optimizer | ADAM |
| Weight Decay | 0.0 |
| Standard-score Observations | online |
| Standard-score Parameters | online |

## A.1 Total Compute

Most calculations were performed on a local computing cluster which offered ten TitanX GPU nodes. We estimate the total computation time, including prototype runs, was approximately 968 GPU hours. We calculated the cost of one run of the benchmark then multiplied it by 10 for this estimation. The computation of the C2ST-ddm on the marginals from existing data was performed on the same cluster but using cpu nodes. According to `mlco2.github.io` this would imply 104.54 kg $CO_2$ at a normal institution; however, our cluster is run exclusively on wind power.

## A.2 SBIBM details

Table 4: Actual bounds of stochastic simulation budget for TMNRE along with number of rounds before the stoppping criterion was reached. Maximum of one round implies that there was no truncation and the method is effectively doing MNRE.

| task | num_simulations | Min simulation count | Max simulation count | Min Rounds | Max Rounds |
|---|---|---|---|---|---|
| gaussian_linear | 1000 | 960 | 1065 | 1 | 3 |
| | 10000 | 9966 | 12004 | 3 | 7 |
| | 100000 | 99702 | 115784 | 3 | 6 |
| gaussian_linear_uniform | 1000 | 952 | 1056 | 1 | 1 |
| | 10000 | 9760 | 10469 | 1 | 4 |
| | 100000 | 100223 | 105468 | 1 | 4 |
| gaussian_mixture | 1000 | 954 | 1072 | 1 | 4 |
| | 10000 | 9902 | 10582 | 2 | 4 |
| | 100000 | 99567 | 105704 | 2 | 3 |
| lotka_volterra | 1000 | 966 | 1051 | 1 | 2 |
| | 10000 | 9824 | 11916 | 1 | 6 |
| | 100000 | 99791 | 128290 | 2 | 5 |
| sir | 1000 | 951 | 1024 | 1 | 5 |
| | 10000 | 9973 | 10128 | 2 | 5 |
| | 100000 | 99611 | 100547 | 2 | 3 |
| slcp | 1000 | 949 | 1050 | 1 | 1 |
| | 10000 | 9901 | 10546 | 1 | 1 |
| | 100000 | 99616 | 104968 | 1 | 2 |
| slcp_distractors | 1000 | 951 | 1035 | 1 | 1 |
| | 10000 | 9931 | 10141 | 1 | 1 |
| | 100000 | 99431 | 100882 | 1 | 1 |
| two_moons | 1000 | 934 | 1056 | 1 | 4 |
| | 10000 | 9941 | 10558 | 1 | 4 |
| | 100000 | 99863 | 104919 | 2 | 3 |

We performed a marginalized version of the SBI benchmark on all tasks from [18], except the Bernoulli Generalized Linear Model. Each task has ten parameters drawn from the corresponding prior. Each of those parameters are pushed through the simulator and those become ten observations with a known ground truth posterior and true generating parameter. For the details of each task we refer the reader to Ref. [18] where they are defined at great length. A summary of some of the details for each of these tasks, and the algorithm applied to them, are contained in the Experiment Table 2.

Although it is not required for TMNRE, we applied the stochastic process from [44] to generate samples from the prior distribution. This led to an estimated number of samples for each task. In general, the number of training samples lied within $\sim 5\%$ of reported value in Figure 1. This was not true for the Lotka-Volterra and Gaussian Linear tasks because some runs had more truncation rounds than expected with every round introducing more samples.

Five runs did not converge with TMNRE, those runs were on Gaussian Linear at 1,000 simulations with observation numbers 4 and 5. Lotka-Volterra had the same problem with 10,000 for observations 2 and 6 and again at 100,000 simulations with observation number 6.

The other methods estimated the joint posterior in some manner while TMNRE targeted the marginals directly. The full list of alternative methods are called REJ-ABC, NLE, NPE, NRE_B, SMC-ABC, SNLE, SNPE, SNRE_B. These methods represent a significant portion of the neural simulation-based inference literature and will not be described in detail here. Please consult Ref. [18].

We defined a summary of the C2ST across the task's marginals by taking the average over same-dimensional marginals and averaged over the observations, see (9). These values are reported for

all methods in Figure 1 where the 95% confidence intervals are computed for the C2ST-ddm over observations, i.e. the variance across marginals in the C2ST-ddm calculation is not carried forward into the reported uncertainty. We found it to be very small compared to the reported values and was unlikely to make a significant difference.

The authors note that data and code was used from SBIBM which can be found on GitHub at https://github.com/sbi-benchmark/sbibm. It is distributed with the MIT license.

**Our method**   TMNRE was trained to learn all one- and two-dimensional likelihood-to-evidence ratios thereby predicting the posterior distribution. Since we applied TMNRE, the algorithm truncated the prior distribution depending on the learned marginal likelihood-to-evidence ratio. We gave a generous maximum of ten rounds but no task used so many. The maximum was seven before the stopping criterion was satisfied. We used the ratio of the constrained prior mass from the current round to the previous round, namely $\beta$ in Algorithm 1, as a stopping criterion and set it to 0.8. The stopping criterion was satisfied after a certain number of rounds details about the maximum and minimum round for every task, at every budget, can be found in Table 4. We applied the heuristic for the simulation budget found in Appendix A.8.

The final estimated likelihood-to-evidence ratio approximates the posterior on the constrained region. Samples were drawn from this posterior using rejection sampling. The samples from these marginals, in the constrained region, are used for the reported C2ST-ddm in Figure 1.

## A.3   Torus details

We use a simulator and prior with a torus shaped posterior to showcase three aspects of TMNRE. The ground truth can be seen on the left in Figure 2. We proceed with the details of the simulator followed but subsections which give the details for every showcase experiment.

If we let $\boldsymbol{\theta}, \boldsymbol{g}(\boldsymbol{\theta}) \in \mathbb{R}^3$ and $\theta_k$ denote the $k$th element of $\boldsymbol{\theta}$, then the simulator for this problem is defined $\boldsymbol{g}(\boldsymbol{\theta}) = (\theta_0, \sqrt{(\theta_0 - a)^2 + (\theta_1 - b)^2}, \theta_2)^T$. The likelihood is defined by an additive noise model, namely $p(\boldsymbol{x} \mid \boldsymbol{\theta}) = \mathcal{N}(\boldsymbol{g}(\boldsymbol{\theta}), \boldsymbol{\Sigma})$ where $a, b$ are constant scalars and $\boldsymbol{\Sigma}$ is a diagonal, positive definite matrix. In our experiments we let $a = 0.6$, $b = 0.8$, and $\boldsymbol{\Sigma} = \text{diag}(0.03^2, 0.005^2, 0.2^2)$. To ensure an approximately torus-shaped posterior, we select a "noiseless" observation of interest $\boldsymbol{x}_0 = \boldsymbol{g}(\boldsymbol{\theta}_0)$ and parameters $\boldsymbol{\theta}_0 = (0.57, 0.8, 1.0)^T$.

### A.3.1   Torus TMNRE and MNRE Metrics

Since we did not have a clear simulation budget during the initial run of TMNRE, we determined the number of simulations in the following round by multiplying the retained simulations by 1.5 and sampling from a Poisson distribution. We started with 5,000 requested samples and up to 10 rounds. In the end that meant we ran Algorithm 1 with the following number of simulations in each round: 4985, 11322, 21127, 32032. The stopping criterion was met in four rounds, before the maximum number of rounds was reached.

A sample visualization of this truncation process is visible in Figure 6. As described in the text, each of these truncated priors were utilized for an ablation study where we estimated the marginal likelihood-to-evidence ratio using either the truncated prior or the true prior. In effect, testing the value of TMNRE versus MNRE. Once the number of simulations were fixed by TMNRE we used exactly the same number of simulations at that stage with MNRE.

### A.3.2   Epsilon Hyperparameter Scan

To determine a useful default value for the cutoff threshold $\epsilon$, we ran TMNRE on the torus simulator, as described above, at 10 different values of epsilon. Namely, $\epsilon_i \in \{10^i : i = -1, \ldots, -10\}$. At every round, the simulator requested approximately 10,000 more simulations than were retained from the previous round. The amount of simulations was determined stochastically, see Appendix A.8. Once the method had hit the stopping criteria, the one- and two-dimensional C2ST-ddm was computed and normalized by the number of simulations required to generate it. The results were plotted by truncation cutoff $\epsilon$ on the right in Figure 2. We determined that $10^{-6}$ minimized the C2ST-ddm approximately as well as the global minima $10^{-4}$ while truncating the prior more conservatively.

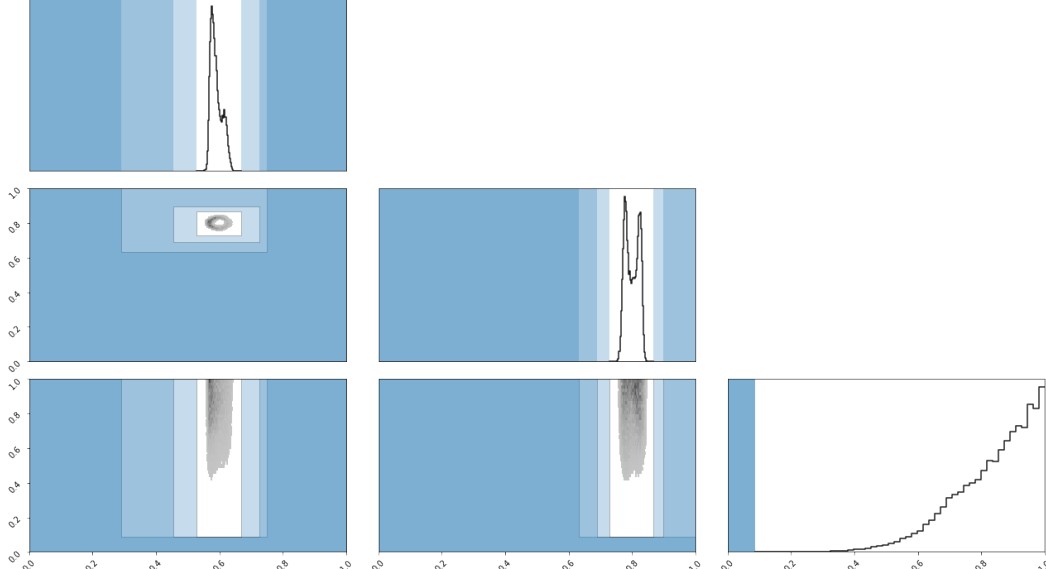

Figure 6: An example of what the truncation process looks like in both one and two dimensions. At each round, the truncated region of the indicator function is denoted in blue. The region is slightly transparent so that the evolution over rounds can be discerned. This plot helps visualize that truncation only occurs at very, very low posterior density. The entire prior region is shown to emphasize that naive sampling results in few samples within the region of interest.

### A.3.3 Empirical Self-Consistency Test

When we do not have a ground truth to compare to, it is important that we can determine whether the nominal credible intervals correspond to the true credible intervals. We propose to do so by comparing the nominal credibility to the empirical credibility. An explanation of how to calculate this performance metric is provided in Appendix B.3.

The authors note, as addressed briefly in the main text, that our consistency test checks the calibration [23] of the posterior to the *truncated* prior, rather than the true prior. This may introduce a bias into the estimation of the self-consistency compared with drawing samples from the truth prior, although we expect it to be small when $\epsilon$ is small. Investigation of the effects are left for future work.

### A.4 Alternative Simulation-based Inference Methods on the Torus

We applied the various SBI techniques to the torus problem for comparison in Figure 7. Just like with MNRE we expected that amortized, non-sequential methods would not have sufficient simulations in the relevant region to accurately model the joint posterior on this task. Similarly, we expected that sequential methods could benefit by focusing samples in the relevant parameter region, thereby learning a more accurate posterior for this piece of data. (Just like TMNRE.) This is what we observe. Our goal in Section 3.2 was to show that truncation offers accuracy with a limited simulation budget, just like sequential methods do.

### A.5 Eggbox details

The eggbox task is well described in the main text. The hyperparameters for NRE, SNRE, and SMNRE are all the defaults as determined by SBI [51]. We implemented SMNRE by creating a custom version of the simulator. We revealed the one or two parameters which were learned sequentially to the SMNRE algorithm while we "baked-in" the uniform prior for the other dimensions, i.e. the simulator sampled from a uniform distribution and simulates a concatenation of the sequentially predicted dimensions with the uniformly predicted dimensions. We expect that SMNRE fails due to its very limited number of simulations. This limitation might seem pathological in this symmetric setting but it is very real in a simulator which defines an unknown posterior that may or may not have symmetry.

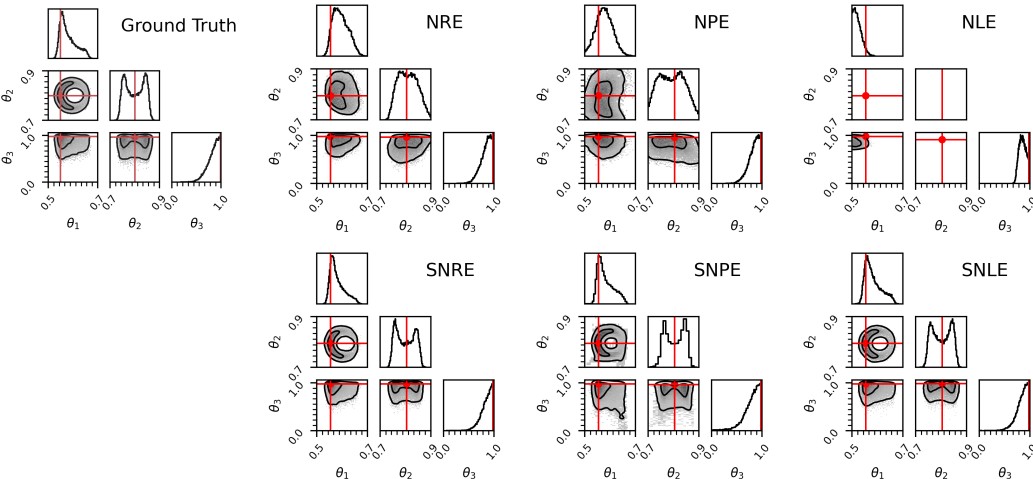

Figure 7: Joint posteriors estimated by various popular SBI methods on the torus problem. When compared to the prior, the narrowness of this posterior renders most untargeted simulations uninformative. Only sequential or truncation-based methods are able to recover accurate estimates at the presented simulation budget of 32032. *Left column*: Torus ground truth from rejection sampling. *Mid-left column*: NRE places mass in the correct region but misses the low density region. SNRE fairly accurately reconstructs the posterior, although slightly narrow in some regions. *Mid-right column*: NPE produces a wide, poorly resolved posterior. SNPE recovers the low density region but introduces incorrect aberrations. *Right column*: NLE fails to accurately place any posterior mass. SNLE accurately reconstructs the posterior.

## A.6 Alternative Simulation-based Inference Methods on the Eggbox

We applied the various SBI techniques to the eggbox problem for comparison in Figure 8. The purpose of this task was to show that multimodal distributions can be challenging for methods which estimate the joint. On this problem we found that to be true for NRE, SNRE, and NPE, see figures 8 and 5. We were unable to draw the necessary samples from SNPE, likely indicating poor performance.

On the other hand, NLE and SNLE seem to work beautifully for this problem, producing accurate looking posteriors. Likelihood estimation methods generally perform very well on benchmark / toy problems presented in the SBI literature. NLE and SNLE are designed to estimate the data distribution using a flow conditioned on parameters. This works well for low dimensions and simple data; however, flows are notoriously difficult to train on higher-dimensional problems due to the necessary computation of the determinant [36] which scales with dimension $d$ like $\mathcal{O}(d^3)$. Evidence of this can be seen in the SLCP-Distractors task in Section 3.1. Introducing extra dimensionality, even when it contains no useful information for inference, significantly reduced the performance of NLE and SNLE. Since this work is intended to create a tool for data with $d \gg 100$, we recommend our method over NLE and SNLE until they can be applied to high dimensional problems.

## A.7 Rotated Eggbox

Truncating with one-dimensional marginals may lead to larger volumes than necessary to contain the posterior mass, when it is not axis-aligned. For example, consider the inefficiencies of truncating a highly correlated Gaussian posterior in this way, see Appendix D. We wanted to create a posterior which was not axis aligned, thereby simulating a truncation scheme which was forced to truncate a much larger volume than necessary to contain the mass of interest. Our next study considers the eggbox problem transformed by a rotation to remove its axis-alignment.

We created a rotated version of the eggbox simulator where the initial simulator $g(\boldsymbol{\theta})$ was replaced by $g(Q^T \boldsymbol{\theta})$. $Q \in \mathbb{R}^{10 \times 10}$ is a rotation matrix which rotates the point $(1, 1, \ldots, 1)^T$ to $(0, 0, \ldots, c)^T$ where $c$ is a positive constant. We had been using a uniform prior defined by the unit cube; however, this region no longer holds the posterior mass. We determined the limits of a hyperrectangle which completely covered the rotated prior in the following manner: Consider each unit vector pointing

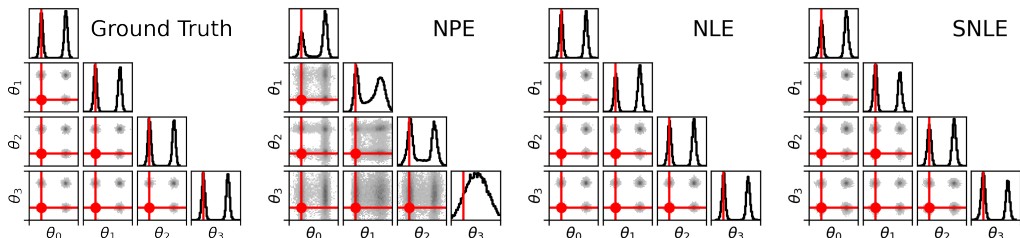

Figure 8: Posteriors from the 10-dim eggbox benchmark with only four parameters shown for clarity. Each method recieved 10k training samples and produced 25k posterior samples. All methods, other than the ground truth, were trained jointly. NPE fails to model the posterior. SNPE training converged, but drawing the posterior samples required well over three days of computation. NLE and SNLE seem to have accurately recovered the ground truth joint distribution with the budget.

to the edge of the unit cube $E = \times_{n=0}^{10}\{0, 1\}$ where $\times$ denotes the Cartesian product. If each of those vectors $v_i$ is rotated by $Q$, $v_i \mapsto Qv_i$, we can then look at the minimum and maximum values projected along each basis vector and set those to be the limits of our new uniform prior. The bounds were $[(-0.924, 0.924), \dots, (0.000, 3.160)]$ which implies a volume 795 times larger than the unrotated prior. We place our new uniform prior over this volume to simulate an inefficient truncation.

Generating samples from the ground truth joint posterior is rather difficult because of the extremely large number of modes and lack of symmetry. To generate samples from the rotated distribution, we simply rotated samples from the original eggbox problem. However, this does not reveal the entire posterior since the simulator is periodic over the prior interval. We first copied the 10,000 samples over all of the hypercubes neighboring the unit cube, yielding 20,470,000 samples, then sub-sampled those and rotated them. More translations were necessary to completely represent the periodic nature of the ground truth in Figure 9, but we quickly ran into memory constraints. The limited, single-set-of-neighboring-hypercubes representation of the rotated eggbox is shown in Figure 9.

For our experiment we trained all SBI methods along with MNRE on this problem using the same hyperparameters as in the original eggbox. No method was able to faithfully represent the ground truth. We believe that the methods failed due to the extremely large volume of the prior rather than the rotation.

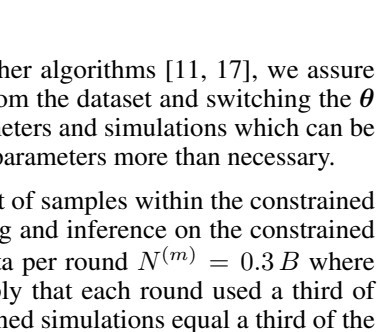

Figure 9: The ground truth of the rotated eggbox posterior estimated by a set of translations and a rotation of samples from the eggbox. The first eight dimensions are symmetric so only two are shown along with the final asymmetric dimension.

### A.8 Simulation budget and dataset generation

First, we note that it is possible to accomplish Algorithm 1 without having to sample new independently drawn parameters by pairing simulations with other parameters. Like several other algorithms [11, 17], we assure the independence of $x$ and $\theta$ by sampling two mini-batches from the dataset and switching the $\theta$ parameters. This produces a pair of independently drawn parameters and simulations which can be used to calculate the loss function efficiently without sampling parameters more than necessary.

Second, we discuss our heuristic for producing a useful amount of samples within the constrained region. We divided the simulation budget between constraining and inference on the constrained region. During the constraining phase, we set the training data per round $N^{(m)} = 0.3\,B$ where $B$ represents the entire simulation budget. This does not imply that each round used a third of the simulation budget, rather the new simulations plus the retained simulations equal a third of the budget. Finally, once the stopping criterion was satisfied, we used the remaining budget within the estimated $\hat{\Gamma}^{\text{rec}}$. We found that this technique created enough simulations during the truncation rounds to estimate $\Gamma^{\text{rec}}$ relatively well while leaving a sizable portion of the simulation budget to be sampled

from the truncated prior. Naturally, we want to sample as much from the truncated prior as possible to reduce simulations in regions of nearly zero posterior density and increase simulator efficiency.

In contrast, sequential methods usually divide their simulation budget evenly across rounds. However, since they do not have a stopping criteria it is natural to divide the simulations that way. We used this technique when training sequential methods.

Third, we discuss the stochastic nature of our sampling technique. Rather than sampling an exact number of parameters and corresponding simulations, we instead sampled from a Poisson distribution centered at the the requested number of samples. In practice, this meant around a 5% difference between the extrema of the actual number of produced simulations and the requested number of simulations.

### A.9  How do we use the likelihood-to-evidence ratio?

**Histograms.**  Domain scientists, in particularly astronomers and astrophysicists, typically consider a visualization of the posterior and draw conclusions based on their problem-specific intuition and the reported uncertainty bounds. By learning the relevant one- and two-dimensional marginal likelihood-to-evidence ratios, our method can generate a visualization of the posterior, namely a corner plot of weighted histograms, by sampling from the prior $\boldsymbol{\vartheta} \sim p(\boldsymbol{\vartheta})$ and sorting the results into bins. Each sample's contribution is weighted according to the learned $\hat{r}(\boldsymbol{x} \mid \boldsymbol{\vartheta})$, creating a posterior histogram.

The histogram facilitates the computation of credible regions. In particular, finding an accurate estimate of the $(100 - \alpha)\%$ highest density credible region is the primary goal of most astronomers.

**Rejection sampling.**  We can use our unnormalized point-wise posterior estimate as the target and the constrained prior as the proposal to generate samples distributed like the posterior via rejection sampling. Let $\tilde{p}_\Gamma(\boldsymbol{\vartheta} \mid \boldsymbol{x}) = \hat{r}_\Gamma(\boldsymbol{x} \mid \boldsymbol{\vartheta}) \mathbb{1}_\Gamma(\boldsymbol{\vartheta}) p(\boldsymbol{\vartheta})$ be our target distribution and $q(\boldsymbol{\vartheta}) = \mathbb{1}_\Gamma(\boldsymbol{\vartheta}) p(\boldsymbol{\vartheta})$ be our proposal distribution. $\mathbb{1}_\Gamma$ denotes an indicator function which is nonzero in constrained region $\Gamma$, $\tilde{p}_\Gamma(\boldsymbol{\vartheta} \mid \boldsymbol{x})$ is the unnormalized posterior, and $\hat{r}_\Gamma$ is the constrained likelihood-to-evidence ratio. Following a modified version of Maximum Likelihood Estimate (MLE)-based rejection sampling [64], we set $M = \hat{r}_\Gamma(\boldsymbol{x} \mid \hat{\boldsymbol{\vartheta}})$ where $\hat{\boldsymbol{\vartheta}} = \arg\max_{\boldsymbol{\vartheta}} \hat{r}_\Gamma(\boldsymbol{x} \mid \boldsymbol{\vartheta})$ is the MLE. We sample parameters from the proposal distribution $\boldsymbol{\vartheta} \sim q(\boldsymbol{\vartheta})$ and accept them with probability $\frac{\tilde{p}_\Gamma(\boldsymbol{\vartheta} \mid \boldsymbol{x})}{Mq(\boldsymbol{\vartheta})} = \frac{\hat{r}_\Gamma(\boldsymbol{x} \mid \boldsymbol{\vartheta})}{\hat{r}_\Gamma(\boldsymbol{x} \mid \hat{\boldsymbol{\vartheta}})}$.

The acceptance probability is tolerable when the parameter space is low dimensional and the constrained prior is not significantly wider than the posterior [65]. For our method, the first condition generally holds but the second is not guaranteed. Despite this potential inefficiency, the parallel proposal and rejection of samples is resolved quickly when the acceptance probability is not vanishingly small. In that case, likelihood-free MCMC [11, 17, 18] becomes unavoidable.

## B  Evaluation metrics

We introduce here the relevant evaluation metrics. These are relevant both to compare our results with the ground-truth (C2ST, KL divergence), as well as for studying desirable statistical properties of the posterior without requiring knowledge of the ground-truth (self-consistency check / empirical credible interval testing / expected coverage testing). Here, C2ST is motivated by its omnipresence in the simulation-based inference literature, to which we want to compare. We additionally introduce KL divergence as a metric that is tractable for the low-dimensional marginal posteriors that are the focus of this paper.

The neural likelihood-free inference reports several performance metrics which do not apply well to our method... Reporting $-\mathbb{E}[\log q(\boldsymbol{\theta}_o \mid \boldsymbol{x}_o)]$ is quite common throughout the literature [11, 13, 14, 15, 17]. Since we learn an unnormalized posterior, we cannot compare our value to other methods. Furthermore it is a poor indicator of performance [18]. Another common technique is to measure the median distance between posterior-predictive samples [13, 14, 17] but this is impossible since we learn a marginalized posterior and cannot sample from the posterior predictive distribution. Maximum Mean Discrepancy (MMD) [66, 67] has been found to be sensitive to choice of hyperparameters [18]. It is in principle possible to apply other alternatives such at the Wasserstein distance [68] using the Sinkhorn-Knopp algorithm [69] but there is not literature precedent.

## B.1 Kullback–Leibler Divergence

Since this paper is primarily interested in determining low dimensional marginal posteriors, it is feasible to estimate the Kullback–Leibler divergence, denoted $D_{KL}$, using samples and comparing the histograms. We've found that this method for approximating the Kullback–Leibler divergence is hyperparameter dependent, namely based off the number of bins. We only reported the Kullback–Leibler divergence for the torus problem and we used 100 bins. This effect implies that only the difference between Kullback–Leibler divergences is relevant.

## B.2 Classifier 2-Sample Test per d-Dimensional Marginal (C2ST-ddm)

The classifier 2-sample test (C2ST) [48, 49] is a performance metric where a classifier is trained to differentiate between samples from the ground truth and approximate posterior. It features an interpretable scale where 1.0 implies that the classifier could distinguish every pair of samples the distributions while 0.5 implies indistinguishably. It is possible to determine where distributions differ using this metric [18]. A classifier with insufficient expressivity yields unreliable results [18, 23, 49].

We define the *C2ST per d-Dimensional Marginal (C2ST-ddm)* test statistic, which reports the average C2ST across every set of $d$-dimensional marginals. Consider two random variables $\boldsymbol{X} \sim P(\boldsymbol{X}), \boldsymbol{Y} \sim Q(\boldsymbol{Y})$ with $\boldsymbol{X}, \boldsymbol{Y} \in \mathbb{R}^D$ and hyperparameter $1 \leq d \leq D$ that represents the marginal dimensionality of interest. Let $(S_P, S_Q) := \left\{ (S_{P_k}, S_{Q_k}) : k \in \{1, 2, \ldots, \binom{D}{d}\} \right\}$ where $S_{P_k} := \{\boldsymbol{x}_k^{(1)}, \ldots, \boldsymbol{x}_k^{(n)}\} \sim P(\boldsymbol{X}_k)$ and $S_{Q_k} := \{\boldsymbol{y}_k^{(1)}, \ldots, \boldsymbol{y}_k^{(n)}\} \sim Q(\boldsymbol{Y}_k)$ are sets of $n$ samples drawn from the $k$th $d$-dimensional marginal of $P$ and $Q$, respectively. Now,

$$\text{C2ST-ddm}(S_P, S_Q) := \frac{1}{K} \sum_{k=1}^{K} \text{C2ST}\left(S_{P_k}, S_{Q_k}\right), \text{ with } K = \binom{D}{d}. \tag{9}$$

For our problem, we let $P_k = p(\boldsymbol{\vartheta}_k \mid \boldsymbol{x}_o)$ and $Q_k = \hat{r}_k(\boldsymbol{x}_o \mid \boldsymbol{\vartheta}_k)p(\boldsymbol{\vartheta}_k)$.

## B.3 Empirical Credible Interval Testing

Evaluating the accuracy of a posterior approximation requires access to the ground-truth and the ability to compute a suitable metric or divergence. While acceptable during benchmarking [18], this is impossible for practitioners because they only have access to the observation $\boldsymbol{x}_o$. Domain scientists depend on sanity checks such as coverage testing and comparison between estimation methods to verify that the reported posterior is accurate. Coverage testing is designed for frequentist confidence intervals; however, we apply a similar technique, known in [23] as expected coverage testing, to test the validity of our credible intervals, empirically.

We report a nominal (100 - $\alpha$)% credible region but the effects of approximation or training might have influenced the contour's shape. Our empirical testing checks whether the nominal contour aligns with the contour ground truth by considering many realizations of $\boldsymbol{x}$ and dividing the number of times the corresponding $\boldsymbol{\theta}$ falls within the nominal credible region by the number of $(\boldsymbol{\theta}, \boldsymbol{x})$s that were tested. When this is the case, the blue line and the orange line intersect in visualizations like Figure 3.

One major advantage of an amortized method for a real-world practitioner is the possibility of quickly performing tests like these. During the training process many parameter-simulation pairs have already been generated, we can use them to check the credible intervals of our method. Note that sequential methods cannot do this without great expense because they would have to retrain their posterior estimator on every tested observation. Furthermore, this test checks the properties of a single amortized estimator; however, testing sequential methods in the same way estimates properties of the sequential training, not a single estimator.

## C   Comparing the truncated marginal likelihood-to-evidence ratio to the truth

### C.1   Exemplary error estimates

We will consider the effect of truncation on a multivariate normal distribution, and discuss various limiting cases. Let us assume that the true posterior has the shape of a multivariate normal distribution with mean zero and covariance matrix $\Sigma$,

$$p(\boldsymbol{\vartheta}|\boldsymbol{x}_o) = \mathcal{N}(\boldsymbol{\vartheta}|0, \Sigma) . \tag{10}$$

This implies that the posterior-to-maximum posterior ratio is given by

$$\frac{p(\boldsymbol{\vartheta}|\boldsymbol{x}_o)}{\max_{\boldsymbol{\vartheta}} p(\boldsymbol{\vartheta}|\boldsymbol{x}_o)} = \exp\left(-\frac{1}{2}\boldsymbol{\vartheta}^T \Sigma^{-1} \boldsymbol{\vartheta}\right) , \tag{11}$$

and that the same ratio for all one-dimensional marginal posteriors is given by

$$\frac{p(\theta_i|\boldsymbol{x}_o)}{\max_{\theta_i} p(\theta_i|\boldsymbol{x}_o)} = \exp\left(-\frac{1}{2}\frac{\theta_i^2}{\Sigma_{ii}}\right) . \tag{12}$$

Let us consider an indicator function $\mathbb{1}_\Gamma$ where $\Gamma$ is defined as in Eq. (6), given some small $\epsilon$. In the case of vanishing parameter correlations, the covariance matrix $\Sigma$ is diagonal, and the posterior factorizes like $p(\boldsymbol{\vartheta}|\boldsymbol{x}_o) = \prod_i p(\theta_i|\boldsymbol{x}_o)$. The truncation procedure can then be considered for each of the one-dimensional marginal posteriors separately: Only parameter regions where $|\theta_i| < \sqrt{-2\Sigma_{ii} \ln \epsilon}$ for all $i$ are included in $\Sigma$. In this case $\epsilon$ directly determines how far into the tails posteriors are correctly reconstructed. Using the error function, one can show that the amount of mass that is removed by the truncation is $\epsilon/\sqrt{-\ln \epsilon}$. This motivates our general estimate of an $\mathcal{O}(\epsilon) \max_{\theta_i} p(\theta_i|\boldsymbol{x}_o)$ effect on the truncated posteriors, where the second factor is accounting for the right dimensionality of the expression.

Let us consider the opposite extreme of a maximally correlated posterior, with a covariance matrix that is given by $\Sigma_{ii} = 1$ and $\Sigma_{ij} = 1 - \xi$ for $i \neq j$, and where $\xi \ll 1$. Again, marginal posteriors are given by Eq. (12). Since the support of the maximally correlated posterior is essentially focused on the line $\theta_1 \simeq \theta_2 \simeq \cdots \simeq \theta_d$, truncations in all directions are identical. As a result, marginal posteriors are affected exactly as in the previous diagonal case.

Finally, let us consider a mildly correlated posterior in two dimensions. In this case, the region $\Gamma$ would be again identified through $|\theta_i| < \sqrt{-2\Sigma_{ii} \ln \epsilon}$ for $i = 1, 2$, but since the posterior does not factorize anymore integrals on the constrained region become non-trivial. However, since only $\mathcal{O}(\epsilon)$ of posterior mass lies outside of $\Gamma$, this implies that only a similarly small mass fraction can be re-distributed in the truncated marginal posteriors $p_\Gamma(\theta_i|\boldsymbol{x}_o)$. This can significantly affect the far low-mass tails of the distribution, with negligible effect on the high mass density regions of the posterior.

### C.2   A general estimate

Let us consider an indicator function defined through Eq. (8), first for a single marginal $\theta_i$. The removed probability mass is then given by

$$\delta M_\epsilon = \int_{\Omega_i} d\theta_i p(\theta_i|\boldsymbol{x}_o) \mathbb{1}\left[p(\theta_i|\boldsymbol{x}_o) < \epsilon \max_{\theta_i} p(\theta_i|\boldsymbol{x}_o)\right] ,$$

where $\mathbb{1}$ denotes an indicator function. An upper bound on the removed probability mass is then given by

$$\delta M_\epsilon < \epsilon \max_{\theta_i} p(\theta_i|\boldsymbol{x}_o) \int_{\Omega_i} d\theta_i \mathbb{1}\left[p(\theta_i|\boldsymbol{x}_o) < \epsilon \max_{\theta_i} p(\theta_i|\boldsymbol{x}_o)\right] ,$$

For a compact $\Omega_i$, this leads to the claimed bound in one dimension. However, also in the case of a larger number of parameters, each truncation would remove at most mass at the level of $\mathcal{O}(\epsilon)$, leading to an overall $\mathcal{O}(\epsilon)$ effect on the estimated posteriors. We emphasize that in the case of priors with non-compact support, a re-parametrization onto priors with compact support can lead to smaller coefficients in front of $\epsilon$.

# D Limitations

We note two kinds of limitations: First, we address limitations when the method works as planned. Second, we address failure modes.

When the posterior distribution is nearly as wide as the prior, we do not gain much by truncating the prior distribution. In this case our method would reduce to MNRE. However, this is rarely the case in physics where the paradigm is to define an uninformative prior distribution across the accepted bounds for a parameter and the posterior will be contained in fractions of that prior's mass.

Another limitation is that the truncation by hyperrectangle is inherently inefficient when the marginals of interest are highly correlated. In that situation, we are interested in a hyperellipse within the constrained hyperrectangle but our current formulation cannot utilize this heuristic. This problem possible to solve by using techniques from Nested Sampling which regularly seeks to efficiently sample from within a certain density contour.

Finally, as mentioned in Section 2.2. Algorithm 1 does not naively allow for sampling from the posterior predictive distribution $p(\boldsymbol{x}' \mid \boldsymbol{x}) := \int_\Omega p(\boldsymbol{x}' \mid \boldsymbol{\theta}) p(\boldsymbol{\theta} \mid \boldsymbol{x}) \, d\boldsymbol{\theta}$ because it only produces the one- and two-dimensional marginals. It remains possible to estimate the joint posterior within the truncated region by simply training another ratio estimator with all parameters. Doing this, using only the simulations necessary for Algorithm 1, may produce an inaccurate joint posterior estimate.

The failure modes are perhaps more obvious. If our initial round of sampling is too sparse, it is possible to incorrectly "miss" a region of high posterior density and cut it out of our analysis. If the initial region is satisfactorily sampled from, this will not occur. To illustrate this point, consider a simulator with a two-dimensional parameter space. If significant amounts of posterior mass are truncated in the $\theta_0$ dimension, the ground truth $\theta_1$-marginal posterior, under the truncated prior, transforms from the intended marginal distribution $p(\theta_1 \mid \boldsymbol{x})$ towards a conditional distribution $p(\theta_1 \mid \boldsymbol{x}, \theta_0^*)$. Where $\theta_0^*$ denotes the center of the truncated region in the $\theta_0$ dimension.

Another failure mode is related to the local amortization that our ratio estimators learn. While they are able to estimate any posterior from a parameter drawn within the truncated prior, it may be that some of the posterior runs into the truncation. This can be identified whenever a posterior equicontour line intersects with the truncation bounds. The caveat is that if an entire separate mode is truncated, this test will not indicate it. In general, we suggest limiting the use of the locally amortized predictions to ones closer to the ground truth generating parameter than to the truncation bounds.

# E Cosmological inference with a simulator

Parameter inference plays a important role in modern cosmology. Here we use a simulator that takes six parameters (specifying the underlying $\Lambda$CDM cosmological model) and returns three lensed angular power spectra $C_\ell^{TT}, C_\ell^{TE}, C_\ell^{EE}$ (where $T$ denotes temperature and $E =$ denotes E-mode polarization) as they would be measured by an idealized Cosmic Microwave Background (CMB) experiment. The likelihood-based approach to inference in this context is provided by popular packages such as `MontePython` [70]. Our simulator is identical to the likelihood that in `MontePython` is called `fake_planck_realistic` [71]. This likelihood is used, often in combination with other likelihoods, to forecast the expected constraining power of future experiments. In this model, the power spectra receive non-stochastic contributions from the cosmological model and the idealized measurement instrument. Stochasticity is implemented in the form of *cosmic variance*, which reflects the fact that for fixed $\ell$, each $C_\ell$ is determined by measuring $2\ell + 1$ modes in the sky. The result is that the collection of $C_\ell$ obeys a Wishart distribution, which at large $\ell$ can be approximated as a multivariate normal distribution. For more details see [72]. Draws from the simulator with and without noise are shown in Fig. 10.

We use this example to study the utility of *marginal* ratio estimation. As such, we do not use multiple rounds of simulation and training. This is in part due to the availability of a tractable likelihood, which allows us to perform a Fisher estimation (i.e. Gaussian approximation) of the expected marginal probability contours. Although the ground-truth posteriors for this inference problem turn out to be slightly non-Gaussian, the Fisher estimation suffices to derive a reasonable region in parameter space for inference. We therefore take a uniform prior with $\theta_d \in \left[\overline{\theta}_d - 5\sigma_d^F, \overline{\theta}_d + 5\sigma_d^F\right]$, where $\overline{\theta}_d$ denotes

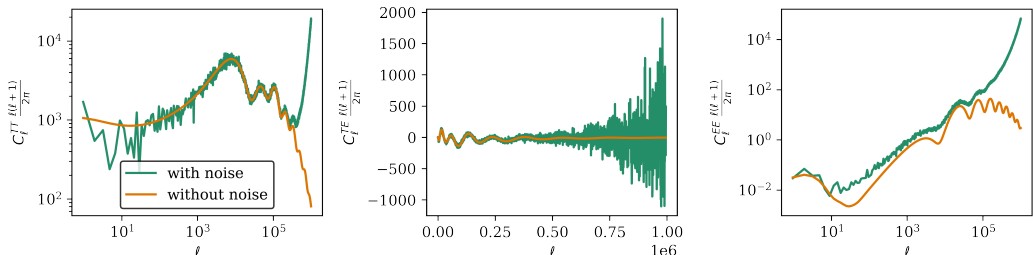

Figure 10: A sample drawn from the CMB simulator, with the cosmological contribution in orange and the noise-added sample in green. The noise model amounts to a non-stochastic contribution from the instrument as well as a stochastic contribution (following a Wishart distribution) corresponding to cosmic variance, in other words the fact that an individual $C_\ell$ is determined by measuring $2\ell + 1$ modes.

the ground truth parameter value and $\sigma_d^F = \sqrt{(F^{-1})_{ii}}$ is the Fisher estimation of the $1\sigma$ region for parameter $i$.

We compare three approaches with 5,000 samples. For comparison, an MCMC analysis of this problem converges after roughly 45,000 accepted samples with an acceptance rate of $\sim 0.3$. We compare MNRE, NRE, and MCMC with a limited number of samples. For MCMC we use a pre-computed covariance matrix for proposal steps, determined by running a chain until convergence. For inference with MNRE and NRE, we use a linear compression layer that takes the concatenated power spectra (each with $\ell \in [2, 2500]$, so that the full data vector has 7497 entries) and outputs 10 features. The same linear compression network is shared between different ratio estimators. In other words, we introduce a shared feature embedding of the data such that the entire neural network has the form

$$f_{\phi,k}(\boldsymbol{x}, \boldsymbol{\vartheta}_k) = g_{\phi_g,k}(\boldsymbol{F}_{\phi_{\boldsymbol{F}}}(\boldsymbol{x}), \boldsymbol{\vartheta}_k) \tag{13}$$

where $\boldsymbol{F}$ is the feature embedding, $k$ represents the index of the marginal-of-interest, $g$ is an MLP, and $\phi$ represents the network weights from both $g$ and $f$. This is appealing computationally but, unlike with $\phi_g$, the weights of the feature embedding are dependent on the loss of every marginal. This is the multi-target training paradigm and can be difficult to tune [73, 74]; however, this is not a problem for us in practice. The hyperparameters are written in Table 5.

The results are shown in Fig. 11. We see that MNRE reproduces the ground-truth 1- and 2-$\sigma$ contours very accurately. On the other hand, NRE results in hardly any constraint on the parameter space, while the limited MCMC run does not have accurate 2-$\sigma$ contours.

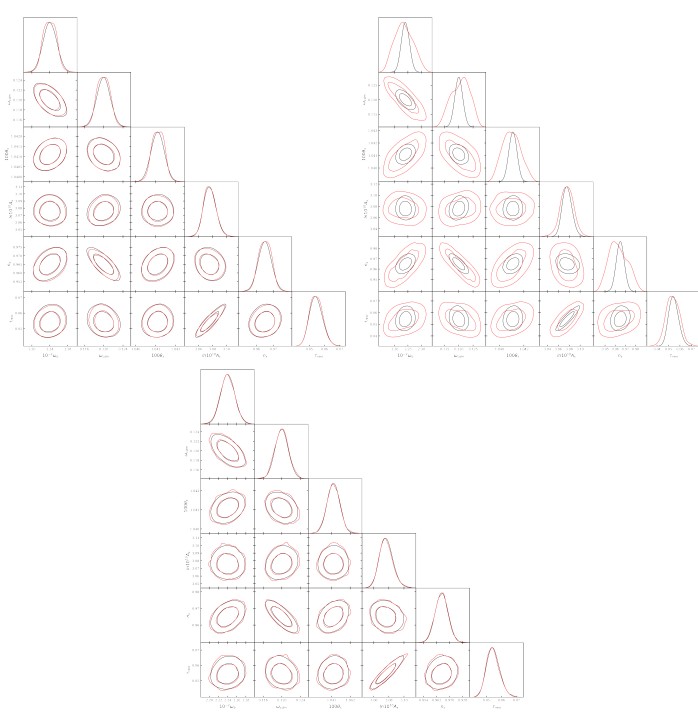

Figure 11: Corner plots for various methods using 5,000 simulations (in red) vs. ground-truth MCMC (45,000 accepted samples with acceptance rate $\sim 0.3$, in black). Top Left: results for MNRE. We see excellent agreement with the ground truth. Top Right: corner plot for NRE. With 5,000 samples, the marginal posteriors are hardly constrained. Bottom: corner plot for MCMC with 5,000 accepted (burn-in removed) samples vs. converged MCMC chain. While the short chain gives accurate $1\sigma$ contours, it does not yield accurate for the $2\sigma$ contours.

Table 5: Physics Example Hyperparameters

| Hyperparameter | Value |
| --- | --- |
| Activation Function | Feature Embedding: None, Ratio Estimator: RELU |
| AMSGRAD | No |
| Architecture | Feature Embedding: One Linear Layer, Ratio Estimator: MLP |
| Batch size | 64 |
| Batch normalization | No |
| Criterion | BCE |
| Dropout | No |
| Early stopping patience | 5 |
| $\epsilon$ | N/A |
| Hidden features | 256 |
| Percent validation | 10% |
| Reduce lr factor | 0.25 |
| Reduce lr patience | 2 |
| Max epochs | 300 |
| Max rounds | N/A |
| Learning rate | 0.001 |
| Learning rate scheduling | Decay on plateau |
| Optimizer | ADAM |
| Weight Decay | 0.0 |
| Z-score observations | online |
| Z-score parameters | online |