# OpenReview forum: "Truncated Marginal Neural Ratio Estimation"
_NeurIPS.cc/2021/Conference — NeurIPS 2021 Poster_

### Official Review · Reviewer_8ziR · 2021-07-10

**Rating:** 6
**Confidence:** 3

**Summary:**

The authors suggest the use of neural ratio estimation to target the marginal posteriors directly, thereby circumventing the more difficult problem of obtaining the full Bayesian posterior.
The method comes with other benefits such as local amortization as well as a sequential truncation technique for targeted inference.
The method is validated on several benchmark examples.


**Ethical Concerns:**

-

**Ethics Review Area:**

["I don’t know"]

**Limitations And Societal Impact:**

Societal Impact has been adequately addressed.

**Main Review:**

__Originality:__ There is something to the idea to circumvent the trouble of full Bayesian inference and target the marginal posteriors directly. The authors also consider modifications that help to improve the performance by truncation. With its aims this paper is a different take on simulation-based inference than previous approaches.

_Some minor comments:_
- References to related work have a recency bias. For example, when writing “...the traditional approach is Approximate Bayesian Computation (ABC) [7]...” the reference [7] is a survey from 2018. However, ABC techniques already emerged in the 1990s. If the authors don’t want to add any more references, then I suggest something like [see e.g. 7 for a reference], in latex `\cite[see e.g.][for a reference]{bibkey}`.

- Inference for simulators goes back at least to the 1980s:
Diggle, Peter J., and Richard J. Gratton. "Monte Carlo methods of inference for implicit statistical models." Journal of the Royal Statistical Society: Series B (Methodological) 46.2 (1984): 193-212.

__Quality:__
While it might be true that many practitioners mostly consider marginal posteriors (be it just because of visualization) I think the paper’s claims are too strong in parts.

For example, Table 1 is very suggestive, effectively showing the properties of the method but indicating that those are _the_ desirable properties of an algorithm. In particular, the “Simulator efficient direct marginals“ are not by necessity a desirable property as I would assume that most Bayesians would prefer to have access to the joint distribution if cost was not an inhibiting factor.

__Clarity:__ The paper is generally clearly written and well organized. I did find some minor points the authors might want to change in a revision to additionally improve the clarity of the paper.

_Some minor comments:_
- May I suggest the authors use `\usepackage[sort]{natbib}`
- I am not a fan of the formula
$$
	f(x | \theta) = \int \delta(x - g(z, \theta)) f(z | \theta) dz
$$
and I feel it adds more confusion than clarity, not least because of its double abuse of notation. First, it abuses notation by using the same symbol for the distribution of $X$, $f( \cdot \mid \theta)$ and for $Z$, also $f( \cdot \mid \theta)$. They would be better references as $f_X$ and $f_Z$. Second, the use of the Dirac delta as a density requires some mathematics that are absent from the rest of the paper and might especially be confusing to non-physicists.  Third, I fail to see any additional content by writing the simulator likelihood in this way, effectively denoting an unknown quantity $f(x \mid \theta)$ as integral over another unknown density $f(z \mid \theta)$ and an additional unknown function $g$.
“The full joint posterior can be unnecessarily informative.” I don’t think that makes sense as a criticism. The additional information of the posterior might not be worth the additional cost afforded, but “unnecessarily informative” is too blunt.
The notation $\{g(\theta, z) : \theta \in \Omega, \forall z\}$ is confusing. The quantifier “for all” ($\forall$) is usually followed by a predicate, but I don’t see what predicate should be satisfied by all $z$.


__Significance:__
Approaches that help reduce the computational burden of simulation-based inference are an important part of research. The current draft of the paper needs some minor improvement before it is ready to be published. However, but I am happy to increase my score if the author can address the points raised here.

__Edit after rebuttal:__
The authors have addressed all my concerns on clarity and have suggested changes that in my opinion improve the quality of the manuscript. I have therefore decided to increase my score towards acceptance.

**Time Spent Reviewing:**

9

---

> ### Author Response · Authors · 2021-08-10
> **Initial comment - 8ziR**
>
> ## Summary
>
> > The authors suggest the use of neural ratio estimation to target the marginal posteriors directly, thereby circumventing the more difficult problem of obtaining the full Bayesian posterior. The method comes with other benefits such as local amortization as well as a sequential truncation technique for targeted inference. The method is validated on several benchmark examples.
>
> The authors would like to thank Reviewer 8ziR for the time donated to review our work and the thoughtful comments. We are happy to address the remaining concerns here, as well as in the camera ready version of the paper.
>
> ## Main Review
> ### Originality
> > There is something to the idea to circumvent the trouble of full Bayesian inference and target the marginal posteriors directly. The authors also consider modifications that help to improve the performance by truncation. With its aims this paper is a different take on simulation-based inference than previous approaches.
> >
> > _Some minor comments:_
> > - References to related work have a recency bias. For example, when writing “...the traditional approach is Approximate Bayesian Computation (ABC) [7]...” the reference [7] is a survey from 2018. However, ABC techniques already emerged in the 1990s. If the authors don’t want to add any more references, then I suggest something like [see e.g. 7 for a reference], in latex `\cite[see e.g.][for a reference]{bibkey}`.
>
> To address the recency bias in ABC references, we included the suggested modification in the first paragraph of the introduction, `\cite[see e.g.][for a reference]{bibkey}` referencing `Scott A Sisson, Yanan Fan, and Mark Beaumont. Handbook of approximate Bayesian computation. CRC Press, 2018.`. In addition, we introduced the following "early work" citations including theoretical `Donald B. Rubin. Bayesianly Justifiable and Relevant Frequency Calculations for the Applied Statistician. The Annals of Statistics, 12(4):1151 – 1172, 1984.380.` and computational  `Simon Tavaré, David J Balding, R C Griffiths, and Peter Donnelly. Inferring Coalescence Times From DNA Sequence Data. Genetics, 145(2):505–518, 02 1997.` contributions after the first mention of the term `Aproximate Bayesian Computation (ABC)` in line 22.
>
> > - Inference for simulators goes back at least to the 1980s: Diggle, Peter J., and Richard J. Gratton. "Monte Carlo methods of inference for implicit statistical models." Journal of the Royal Statistical Society: Series B (Methodological) 46.2 (1984): 193-212.
>
> We have included a reference to `Diggle and Gratton` as a part of our preamble regarding the history of inference for simulators and emphasized the existing body of work on this subject in the second paragraph of our introduction. We changed line 23 to `Simulation-based inference (\SBI) is closely connected to \ABC\ and has been an open research topic since as early as the 1980s \cite{diggle1984monte}.`.
>
> ### Quality
> > While it might be true that many practitioners mostly consider marginal posteriors (be it just because of visualization) I think the paper’s claims are too strong in parts.
>
> We want to acknowledge the reviewer's critique and agree that our claims can safely be blunted while clarifying the overall message. We have made several changes to emphasize that TMNRE comes with limitations, particularly its lack of access to the joint posterior while offering insight into problems where simulation cost is a major bottleneck. We summarize some of the changes below.
>
> The authors rephrased the "Marginal posteriors instead of the joint" paragraph in the introduction to further specify the use-case of our algorithm. The new text reads, `Marginal posteriors instead of the joint. Scientific insight is often based on a low dimensional marginalization of the posterior with nuisance parameters removed \cite{aghanim2020planck}. The additional information of the full joint posterior might not be worth the additional cost afforded. Targeting marginals directly, by estimating only the marginal posterior for the parameters of interest, is simpler and sufficient for many scientific, parameter estimation, and bounding purposes.`
>
> where `\cite{aghanim2020planck}` means Aghanim, Nabila, et al. "Planck 2018 results-VI. Cosmological parameters." _Astronomy & Astrophysics_ 641 (2020): A6.
>
> We added further acknowledgment of the limitations of our work, in particular, regarding the posterior predictive, in a new paragraph between lines 182 and 183 which reads as follows, `Algorithm~\ref{alg:tmnre} estimates the necessary marginal posteriors for corner plot visualization. We demonstrate the cost-effectiveness of this algorithm in Section~\ref{sec:experiments}. An important limitation of Algorithm~\ref{alg:tmnre} is the inaccessibility of the posterior predictive distribution. This limitation is mitigated by training a ratio estimator on all parameters within the truncated region; however, producing an accurate joint estimate may come with (sometimes significant) additional simulation costs.`
>
> Although it was mentioned in the discussion section, this particular limitation was not represented in the limitation section of our Appendix. We have now included it there. It reads as follows, `Finally, as mentioned in Section~\ref{sec:tmnre}. Algorithm~\ref{alg:tmnre} does not naively allow for sampling from the posterior predictive distribution $p(\bx' \mid \bx) \coloneqq \int_\Omega p(\bx' \mid \btheta) p(\btheta \mid \bx) \, d\btheta$ because it only produces the one- and two-dimensional marginals. It remains possible to estimate the joint posterior within the truncated region by simply training another ratio estimator with all parameters. Doing this, using only the simulations necessary for Algorithm~\ref{alg:tmnre}, may produce an inaccurate joint posterior estimate.`
>
> > For example, Table 1 is very suggestive, effectively showing the properties of the method but indicating that those are _the_ desirable properties of an algorithm. In particular, the “Simulator efficient direct marginals“ are not by necessity a desirable property as I would assume that most Bayesians would prefer to have access to the joint distribution if cost was not an inhibiting factor.
>
> Specifically regarding Table 1, we agree with the reviewer's assessment and have changed the caption accordingly: `Comparison of \SBI\ methods, including our proposed \TMNRE, along with select properties. Properties listed are intended to showcase \TMNRE\ and do not necessarily reflect the most desirable properties in every inference setting. For example, if cost were not a inhibiting factor a tractable joint distribution may be more appealing than targeting marginals directly. Similarly, a fully amortized posterior estimate is more flexible than a targeted one but remains, often, prohibitively expensive.`
>
> We refer to Table 1 on lines 59 and 186 in the text to emphasize the properties of our algorithm specifically. In the paragraph beginning on line 186, we introduced a sentence clarifying that TMNRE's properties have a favorable cost-benefit ratio.
>
> `\TMNRE's properties provide a favorable cost-benefit ratio--yielding marginal insight into the posterior without paying the price of accessing the joint posterior. That price is often inhibiting for expensive simulators.`
>
> ### Clarity
> >The paper is generally clearly written and well organized. I did find some minor points the authors might want to change in a revision to additionally improve the clarity of the paper.
> >
> > _Some minor comments:_
> >  - May I suggest the authors use `\usepackage[sort]{natbib}`
>
> We agree with the suggestion. We applied it in the camera ready version, we use the `sort` command when rendering `natbib`.
>
> > I am not a fan of the formula ... and I feel it adds more confusion than clarity, not least because of its double abuse of notation. ... effectively denoting an unknown quantity $f(x \mid \theta)$ as integral over another unknown density $f(z \mid \theta)$ and an additional unknown function $g$.
>
> We decided to remove the sentence which included an integral over a delta function from the paper completely. As Reviewer 8ziR says, it does not introduce much insight while it does introduce other concerns. Since the description of an intractable likelihood is included throughout the paragraph and paper, the sentence was removed without a loss of clarity for the reader. We modified the following sentence, in lines 104 and 105, which now reads: `The joint posterior is given via Bayes' rule as $p(\theta \mid x) = p(x \mid \theta)p(\theta)/p(x)$, where $p(x \mid \theta)$ is the intractable likelihood (or implicit distribution~\cite{diggle1984monte, hartig2011statistical}) and $p(x)$ is the evidence.`.
>
> where `\cite{hartig2011statistical}` means Hartig, Florian, et al. "Statistical inference for stochastic simulation models–theory and application." _Ecology letters_ 14.8 (2011): 816-827.
>
> > “The full joint posterior can be unnecessarily informative.” I don’t think that makes sense as a criticism. The additional information of the posterior might not be worth the additional cost afforded, but “unnecessarily informative” is too blunt.
>
> We agree with the reviewer's statement. We replaced this sentence in lines 38 and 39 by  `The additional information of the full joint posterior might not be worth the additional cost afforded.`
>
> > The notation $g(\theta, z):\theta, \forall z$  is confusing. The quantifier “for all” ($\forall$) is usually followed by a predicate, but I don’t see what predicate should be satisfied by all $z$.
>
> We agree that writing $\forall z$ is not clear. When introducing $z$, we now write "stochastic latent state $z \in \mathbb{R}^{N_z}, N_z \in \mathbb{N}$," and then we replaced the relevant statement {$\{ g(\theta, z) | \theta \in \Omega , \forall z \}$} by  {$\{ g(\theta, z)  |  \theta \in \Omega , z \in \mathbb{R}^{N_{z}} \}$}.

---

> > ### Author Response · Authors · 2021-08-10
> > **Closing Statement**
> >
> > ## Closing Statement
> >
> > We thank Reviewer 8ziR for the consideration of our work and the effort spent to peer review it. The reviewer raised important points regarding additional references that ought to be added, where our claims were too strong, and actionable changes about clarity. Given our best efforts to answer all the elements raised in the review, we hope we have now resolved the concerns and that the initial rating will be revised accordingly as suggested in the "Significance" section. If there are remaining or new concerns, we would be glad to discuss them with you.

---

### Official Review · Reviewer_BG8g · 2021-07-12

**Rating:** 6
**Confidence:** 4

**Summary:**

This paper presents a method for performing likelihood-free inference in parametric simulators.  The method presented is thematically similar to, and builds upon, a recent tranche of work using neural function approximators to ameliorate the inherent difficulties in likelihood-free settings.  The method takes the interesting and unique approach of modelling an arbitrary set of many smaller marginals of the full posterior, instead of directly targeting the full posterior.  These marginals are estimated by using neural ratio estimation between the target marginal posterior and the full marginal.  The prior is then truncated (to a hyperrectangle smaller than the support of the original prior in each dimension) to concentrate simulations into regions of significant posterior mass.  These estimators (the ratio estimators and the truncated prior) can then be used to establish an estimate of the posterior distribution.  The method is evaluated on reasonably standard tasks, and an interesting cosmology example (although this was inexplicably demoted to the supplementary materials).

**Ethical Concerns:**

No ethical concerns of note.

**Limitations And Societal Impact:**

No societal impacts of note.

**Main Review:**

### Summary+

The authors posit that modelling many lower-dimensional marginal posterior distributions is a fundamentally easier task than modelling a high-dimensional posterior distribution.  This seems like an interesting and intuitive design choice.  It also seems to be a relatively novel formulation in this context (see comments below).  I can believe that learning multiple cheap classifiers/ratio estimators is computationally streamlined, and defines a fundamentally easier inference task.

The prior art is well surveyed (although the nuts and bolts of the prior art are not explored and hence some familiarity is required).  The problem formulation paragraphs (S2.P1,2) are compact and neatly introduces the problem domain, and indicates the solution path that the authors will take.

The algorithm block is beautifully presented and makes the method very easy to understand.  I believe that I could implement the method using pretty much just the algorithm block.  Chapeau.

The authors benchmark their method on the LFI benchmark dataset, as well as applying their method to two further examples.  The authors TMNRE method performs slightly worse on the benchmark tasks compared to the baseline methods.  A major point in the SBI baselines paper is that the choice of metric is significant, and so I would have liked to have seen these methods compared using a slightly broader range of metrics than a single metric.

I broadly like the submission, although I am not entirely convinced by it and its significance  (see comments below).  I have erred on the side of weak acceptance because I believe the work is sound, the paper is well put together and executed, and it has been well evaluated and the results thoroughly analysed.  The paper is generally well written, and is actually quite enjoyable to read.  The authors also release code, for which they are to be commended for.  However, there is no theoretical innovation or particularly inspiring or transferrable methodological innovation.  The performance on the standard benchmarks is not outstanding, and the baselines were not fully tested on the other experiments, making meaningful comparison difficult.  Finally, the extent to which this method advances our understanding of LFI more broadly is not especially clear to me.  For me, these points limit the reach and impact of this paper.  I would, however, consider upgrading my score if my concerns were successfully allayed.


### Major Comments:

1 - Can the authors comment on the fidelity of modelling a posterior distribution as a collection of marginals?  I am not convinced that this would transfer well to distributions with complex and richly structured posteriors.  I also do not agree with the author's assertion that a joint distribution can be “unnecessarily informative”.  I would rather “pay more” to recover the full joint, and then marginalize post-hoc, as opposed to recovering a simplified representation from which I cannot recover the full representation.  When targeting the full joint, I believe this method just reduces to NRE+truncation.  NRE is known to perform relatively poorly compared to more sophisticated methods, although the truncation _may_ help somewhat.

2 - There are two methods presented, MNRE and TMNRE, correct?  Why wasn’t MNRE tested in Figure 1?  It also strikes me that the torus model and the eggbox model were well-chosen experiments for this method -- the posterior does basically factorize as a product of marginals with well-concentrated mass, and hence I am not especially surprised that these methods perform well.  I would be interested to see how well something like SNL performs on the torus model.  I suspect it would perform exceptionally well, and provides tractable marginals as stipulated by the authors, and hence I do not see a major contribution from this method there.

3 - Truncating the prior is a reasonably low-overhead method for increasing sample efficiency.  However, for the eggbox example, surely this truncation method leads to the hypervolume retained having a much larger (hyper)volume than one would desire?  This would drive down sample efficiency, especially as regions with posterior mass are driven towards the edge of this volume.  This will _dramatically_ reduce sample efficiency in high dimensions, in a way that more sophisticated sequential methods are not affected.  Furthermore, if the axis were rotated through 45 degrees (in all dimensions), then the volume of the hyper rectangle increases by a factor of 2^5=32, and hence the truncation will be a factor of 32x less efficient.  This seems to be a major drawback to me.

4 - It seems to me that truncation may be a step back from the SNL-esque methods.  I would prefer to see something along the lines of {SMC, MCMC}-MNRE, where instead of truncating the prior, a discrete/atomic representation is used.  This would allow computation to be concentrated more efficiently, would relax the dependence on axis-aligned truncations, and prevent regions of parameter space from being aggressively deleted early in training.  I realise this is not the paper that has been written, but it is a sticking point in my head that a set of strictly contracting axis-aligned truncations is just about the least flexible approach that could have been taken.  You allude to this in the discussion (Lines 349-352).  Do you have any further comments on this?  This seems like the biggest weakness of the method.  (I suppose you want to write Canonical TMNRE next…!)

5 - I do not understand these “consistency checks through local amortization”.  I figure you are suggesting that through the use of synthetic data we can test the inference result by recovering the ground truth.  If this is the case, this seems trivial/generally true to me and does not seem to be a real plus-point of your method in particular.  The real issues with methods such as this arise from model-mismatch in the generative model, and hence there are no real tests for this.  I think this claim is overstated, and unless I’ve gotten something wrong, I would rather see this removed.  (It could be referenced in passing as a sanity check or for debugging -- but nothing more).  I could not follow Section 3.3 as a result.

6 - According to Algorithm 1, the data points aren’t drawn from the prior, they are drawn from the truncated prior, and so is the ratio estimator targeting the correct quantity?  This may just be a typo, i.e. Line 5 and Line 7 are identical (modulo a dash), whereas I believe Line 7 should drop the indicator term.

7 - Using the C2ST score seems a little biased to me.  Since the other methods are computing full posteriors, and yours is only computing a single marginal,  performing a two-sample test comparing yourself to the full posterior actually seems a little harsh on your own method -- even _incorrect_ as a metric!  I suppose this shows your method is working admirably, but also shows that maybe the solutions just happen to admit a factorized posterior (and hence are amenable to your method).  Either way, I find using this as the only performance metric a bit of a head-scratcher.  As mentioned above, I would like to see a more diverse suite of performance metrics examined.

8 - You recover N^2 marginal pairs, and N marginals.  The N^2 + N marginals that you recover will all be dramatically different as a result of marginalizing over N-2 other variables.  Practically speaking, how do you select which distribution to draw from for a particular distribution?  Moreover, how do you actually go about generating a sample from the joint...?


### Other Comments:

A - Are the authors able to comment on the difference between their method and that of Jeffrey & Wandelt?  I believe that the work is sufficiently different to be easily classified as novel throughout, but I think a more substantiated explanation of any similarities and differences would be useful.

B - Does truncating the prior lead to sections of parameter space being permanently deleted?  I don’t think the algorithm could recover if a section with mass were not covered.  I realise this is true for most sampling algorithms, and Fig 4 suggests that the distribution starts broad and then narrows.  But it strikes me as though it could be particularly dangerous in this setting if there is categorically no way to expand the distribution.  Maybe consider adding some discussion of entropy regularization to the prior to the conclusion.

C - There is a relatively substantial limitations section in the supplementary materials.  I would prefer it if some of these limitations were brought up and explored in the main text.

D - The cosmology experiment is by far the most interesting of the experiments.  I do not understand why it was demoted to the supplementary materials.

E - There also seems to be a missed opportunity using the marginals recovered to establish independencies or conditional independencies in the posterior, where this method explicitly enforces the efficiency gains that these independences can provide.

F - An interesting point that is not strongly emphasised is that the same samples can be (re-)used to learn all of the ratio estimators.

G - I am also a little confused about the “local amortization”.  This method is inherently sequential and conditioned on a single $x_o$, and so there is no real amortization going on.  Around Line 190, you say that you train “locally amortized posteriors that are valid for parameters $\theta \in \Gamma^{(m)}$”.  Are suggesting that the function approximators are going to be someway accurate/informative over the entire domain of $\Gamma^{(m)}$?  This should be explained more clearly if this is the case.


### Minor/Typographical Comments:

- Inline or wrapped figures should be avoided.  They are bad for readability and flow.  Figures should really be floated to the top (or bottom at a push) of the page.
- I don’t understand the sentence beginning Line 30: “With existing…”.
- Line 129: I don’t understand how a likelihood ratio estimator can be overconfident...!
- Line 147: Incorrectly rendered characters (I only see boxes).
- Some of the font sizes in the figures are TINY.
- I don’t really understand what Figures 4(b-d) are showing.
- Figure 6 does a much better job of showing the truncation compared to Figure 4a.
- The discussion is very long and doesn’t add much (partly because the rest of the paper is pretty well written).  This could be cut down and used for the cosmology example instead, or, for Figure 6.
- The societal impact statement is too long.  I think it is safe to say here: “There are no societal impacts.” (as is explicitly permitted in the submission instructions).


**Time Spent Reviewing:**

5

---

> ### Author Response · Authors · 2021-08-10
> **Initial comment - BG8g**
>
> ## Summary
> > This paper presents a method for performing likelihood-free inference in parametric simulators. The method presented is thematically similar to, and builds upon, a recent tranche of work using neural function approximators to ameliorate the inherent difficulties in likelihood-free settings. The method takes the interesting and unique approach of modelling an arbitrary set of many smaller marginals of the full posterior, instead of directly targeting the full posterior. These marginals are estimated by using neural ratio estimation between the target marginal posterior and the full marginal. The prior is then truncated (to a hyperrectangle smaller than the support of the original prior in each dimension) to concentrate simulations into regions of significant posterior mass. These estimators (the ratio estimators and the truncated prior) can then be used to establish an estimate of the posterior distribution. The method is evaluated on reasonably standard tasks, and an interesting cosmology example (although this was inexplicably demoted to the supplementary materials).
>
> We are gracious for the overall positive assessment of Reviewer BG8g, and happily address the remaining concerns here, as well as in the camera ready version of the paper.
>
> We are proud of the method's performance on the cosmology example, believe it acts as an illuminating example to convince the community that marginals can be enough for science applications, and seriously considered putting it in the main text. Although the cosmology example demonstrates how the algorithm might be used by practitioners, it may be less instructive about the behavior of TMNRE than our simpler examples. We will update the paper's discussion, and depending on page limits, we will prioritize putting it in the main text (along with clarification about empirical consistency checks and limitations of the method). At the very least, we will make a more visible reference its placement within the supplemental material. We also now reference cosmology experimental data to motivate the scientific usecase of marginal posterior estimates.
>
>
> ## Main Review
> ### Summary+
> > The authors posit that modelling many lower-dimensional marginal posterior distributions is a fundamentally easier task than modelling a high-dimensional posterior distribution. This seems like an interesting and intuitive design choice. It also seems to be a relatively novel formulation in this context (see comments below). I can believe that learning multiple cheap classifiers/ratio estimators is computationally streamlined, and defines a fundamentally easier inference task.
> >
> > The prior art is well surveyed (...). The problem formulation paragraphs (S2.P1,2) are compact and neatly introduces the problem domain, and indicates the solution path that the authors will take.
> >
> > The algorithm block is beautifully presented and makes the method very easy to understand. I believe that I could implement the method using pretty much just the algorithm block. Chapeau.
> >
> > The authors benchmark their method on the LFI benchmark dataset, as well as applying their method to two further examples. The authors TMNRE method performs slightly worse on the benchmark tasks compared to the baseline methods. A major point in the SBI baselines paper is that the choice of metric is significant, and so I would have liked to have seen these methods compared using a slightly broader range of metrics than a single metric.
>
> The goal of our proposed algorithm is to enable reliable parameter inference in situations where per-simulation costs are very high, and joint posteriors are highly complex and too costly to model.  In these settings, the ground truth is generally not known and not available to assess the performance of an inference task.  For that reason, SBI metrics like C2ST which require the ground-truth to be known are not applicable to many usecases that we target with our algorithm.  In cases where the ground truth is not known, the local amortization properties of our algorithm are intended to be used to assess the quality of the posteriors (more on that below).
>
> To clarify this point, we added a sentence at the beginning of the experiments section in line 196 which reads, `These experiments compare algorithms using performance metrics which access the ground truth posterior. For practitioners who normally cannot access the ground truth, \TMNRE\ offers an empirical consistency check (see Section~\ref{sec:empirical_tests}) to assess the quality of the estimated posterior. Such a check is impractical for sequential methods and is one of the primary practical applications of \TMNRE.`.
>
> Consider a concrete example: The astrophysical gamma-ray emission observed from the Milky Way is compromised of tens of thousands of resolved and unresolved point-like sources [0] (mostly neutron stars and supernova remnants). One might be interested in the population properties of these sources (e.g. their spatial distribution, and their luminosity function), which can be modeled as a Bayesian hierarchical model. This model would have a dozen or so population-level parameters, and thousands of parameters that describe individual source positions and luminosities.  A typical challenge is to derive marginal posteriors for the population level parameters. This problem is intractable for likelihood-based techniques (for instance permutation invariance leads to exponentially many modes). In the context of our algorithm, source parameters would be marginalized by treating them as simulator noise, while the population-level parameters could be inferred as a combination of low-dimensional posteriors. Each run of the simulator might require from seconds to hours of computational time, depending on the amount of model detail, and the ground truth will be in general not accessible. These technical challenges are very typical for analyses in physics and astronomy.
>
> That said, we completely agree that assessing the performance of our algorithm with standard SBI metrics is important for comparison with other algorithms and existing literature.  Paper [1] advocates for the use of the Classifier Two-Sample Test which we utilize, with a slight modification (see section B.2), in our quantitative analysis.  It would be an interesting study to evaluate performance with other (applicable to marginals) metrics from [1] and alternatives such as the Wasserstein distance (see line 731).
>
> [0] Fermi-LAT collaboration. Fermi Large Area Telescope Fourth Source Catalog. https://arxiv.org/abs/1902.10045
> [1] Lueckmann et. al, "Benchmarking Simulation-Based Inference", AISTATS 2021
>
> > I broadly like the submission, although I am not entirely convinced by it and its significance (see comments below). I have erred on the side of weak acceptance because I believe the work is sound, the paper is well put together and executed, and it has been well evaluated and the results thoroughly analysed. The paper is generally well written, and is actually quite enjoyable to read. The authors also release code, for which they are to be commended for. However, there is no theoretical innovation or particularly inspiring or transferrable methodological innovation.
>
> Thank you for such a positive assessment of the paper's quality! We believe truncation is appealing and innovative both due to its simplicity and its transferability. In particular, the truncation method may be trivially applied to other simulation-based inference techniques such as Neural Posterior Estimation (NPE) (see [2] for reference) or Neural Likelihood Estimation (NLE) [3]. Although in both applications there are subtleties due to the use of normalizing flows, rather than ratio estimation, which require further study.
>
> We added a clarification on line 52 reading, `As a basis, we adopt likelihood-to-evidence estimation proposed in~\cite{Hermans2019}, although our truncation scheme is applicable to other neural simulation-based inference methods \cite{papamakarios2019sequential, Durkan2020}.`
>
>  [2] Durkan, et. al. On Contrastive Learning for Likelihood-free Inference
>  [3] Papamakarios, et. al. Sequential Neural Likelihood: Fast Likelihood-free Inference with Autoregressive Flows
>
> > The performance on the standard benchmarks is not outstanding, and the baselines were not fully tested on the other experiments, making meaningful comparison difficult.
>
> The reviewer raises a good point that NLE, NPE, etc. were not tested on the eggbox or torus experiments. Reviewer opr4 mentioned the same point. The experiments are currently underway and we will provide results by Thursday. At which point we will introduce those results into the paper's supplemental materials. Predictions / preliminary results about how the techniques will perform are written below addressing point 2.
>
> > Finally, the extent to which this method advances our understanding of LFI more broadly is not especially clear to me.
>
> To our knowledge, our paper is the first systematic study targeting the marginal posterior directly by means of a simulation-based method--a technique made possible by simulation-based inference and thus previously unexplored (Jeffrey&Wandelt also discuss this to some degree, more below). We show that marginal posteriors can be more effectively learned than joined ones in terms of simulation costs.  Further, the truncation scheme is broadly applicable, yet had not previously been used in a neural setting. (Although as Reviewer opr4 noted, truncation has been utilized in ABC since at least 2010, and we added a corresponding remark to the text.)
>
> > For me, these points limit the reach and impact of this paper. I would, however, consider upgrading my score if my concerns were successfully allayed.
>
> We hope that we can convince the referee about the value of our algorithm and how it extends the usability of sbi methods, especially to cases faced by practitioners.

---

> > ### Author Response · Authors · 2021-08-10
> > **Major comments 1-3**
> >
> > ### Major Comments
> > > 1 - Can the authors comment on the fidelity of modelling a posterior distribution as a collection of marginals? I am not convinced that this would transfer well to distributions with complex and richly structured posteriors.
> >
> > Low-dimensional marginals often have a simple structure due to their low-dimensional nature and the down-projection of overlapping joint probability density. We aim to exploit this in order to reduce simulator costs. In our point source example above, the marginal posteriors of population parameters are often nearly Gaussian, whereas the full joint posterior for all parameters would have rather complex structure.
> >
> > In reference to a relevant point below, we would like to emphasize that our method targets a marginal posterior of interest and does not intend to estimate the joint as a product of marginals. We completely agree with the reviewer that this approach would offer low fidelity with respect to the joint posterior because parameter correlations would be ignored. In our method, within any estimated marginal, there can be arbitrary correlation between the remaining parameters. To clarify this point we adjusted the wording in the "Marginal posteriors instead of the joint" paragraph on line 37.
> >
> > Finally, the fidelity of the obtained marginal posteriors, the simulation efficiency in comparison with sequential techniques, etc. is unfortunately very problem specific, but we tried to further elucidate this with examples in the discussion around point 3&4 below.
> >
> > >  I also do not agree with the author's assertion that a joint distribution can be “unnecessarily informative”. I would rather “pay more” to recover the full joint, and then marginalize post-hoc, as opposed to recovering a simplified representation from which I cannot recover the full representation. When targeting the full joint, I believe this method just reduces to NRE+truncation. NRE is known to perform relatively poorly compared to more sophisticated methods, although the truncation _may_ help somewhat.
> >
> > The authors agree that the term "unnecessarily informative" was misused. We have altered lines 38 and 39 to read as follows, `The additional information of the full joint posterior might not be worth the additional cost afforded.`.  Recovering the full joint, then marginalizing post-hoc, offers a significantly richer understanding of the posterior (and access to the posterior predictive) than estimating it marginally.  Our algorithm shines when the joint is intractable (because of high per-simulation costs and a highly complex joint posterior, see the gamma-ray example above), which are very prevalent in physics and astronomy applications. We rephrased "Our Contribution" to specify this in line 50, `We propose an algorithm that simultaneously achieves all three of the above aspects: Truncated Marginal Neural Ratio Estimation (\TMNRE). It approximates the marginal likelihood-to-evidence ratio in a sequence of rounds and shines when the joint posterior is prohibitively costly.`.
> >
> > Indeed the algorithm reduces to NRE with truncation when the full joint is the target. We chose not to use (S)NLE as a basis because it requires modeling the likelihood which may be very complex for scientific usecases. Using (S)NPE as a basis may be an option for future work, although we found (S)NPE struggled to run with the eggbox example. Since NRE is a supervised method, it is possible to more easily assess whether the method is overfitting, preventing truncation of regions which ought not be truncated.
> >
> > > 2 - There are two methods presented, MNRE and TMNRE, correct? Why wasn’t MNRE tested in Figure 1? It also strikes me that the torus model and the eggbox model were well-chosen experiments for this method -- the posterior does basically factorize as a product of marginals with well-concentrated mass, and hence I am not especially surprised that these methods perform well.
> >
> > The authors chose not to test MNRE directly in Figure 1 because MNRE is merely TMNRE without truncation. MNRE was introduced as an ablated version of TMNRE to verify that estimating marginals is indeed simpler on the eggbox problem, which does not use truncation. We advocate for the use of TMNRE generally, rather than MNRE.
> >
> > Our goal with the eggbox example was to highlight the advantages of estimating marginals rather than the joint, and with the torus example to highlight the advantages of the truncation scheme.  The eggbox example does not use truncation since the posteriors are extended over the entire prior range; hence we refer to MNRE in this context.  The eggbox posterior is indeed fully factorizable, a choice that we made in order to obtain visually sharp marginal posteriors (bi-model for 1-dim and with 4 modes for 2-dim marginals). The torus posterior does not factorize into one-dimensional marginals. The eggbox has a long history in nested sampling literature [4] and has simple to understand marginals.
> >
> > > I would be interested to see how well something like SNL performs on the torus model. I suspect it would perform exceptionally well, and provides tractable marginals as stipulated by the authors, and hence I do not see a major contribution from this method there.
> >
> > We agree with the referee that exploring other inference algorithms would be helpful for comparison with our approach, and are now applying SNLE, SNPE, NLE, and NPE to the torus and eggbox problems and will include those figures in the paper's supplemental materials.
> >
> > On the torus problem, given our fixed simulation budget, the authors predict that sequential methods will perform well while non-sequential methods will not receive the simulations necessary to accurately represent the posterior, as seen with MNRE in Figure 2.
> >
> > The eggbox experiment with SNLE, NLE and NPE is already complete (still waiting on SNPE). NPE fails to accurately model the eggbox at the simulation budget, meanwhile SNLE and NLE model the posterior very accurately. This result will be included along with an explanation in a supplemental section with a reference in the main text. The text will be provided here when all experiments are complete.
> >
> > To briefly reply... NPE cannot handle this joint distribution and would perhaps be better treated marginally, like we do with TMNRE. NLE and SNLE's performance indicates the effectiveness of simulation-based likelihood estimation, even on tough problems like the eggbox. Despite the strong performance, likelihood estimation requires fitting a flow to the data distribution which is difficult when the data is complicated, as it is by scientific practitioners who are our intended audience. An example would be the data from the cosmology simulator. Furthermore this effect has been documented in the SLCP Distractors example in [1] where likelihood estimation was confounded by extraneous information in the data.
> >
> > We emphasize that the using the truncation scheme rather than sequential methods has the advantage (a) of local amortization for quality tests in absence of access to the ground-truth (see below) and (b) that various marginals can be trained on the same truncated training data. TMNRE expands the applicability of simulation-based inference generally without sever performance degradation.
> >
> > [4] Feroz, et. al., MultiNest: an efficient and robust Bayesian inference tool for cosmology and particle physics
> >
> > > 3 - Truncating the prior is a reasonably low-overhead method for increasing sample efficiency. However, for the eggbox example, surely this truncation method leads to the hypervolume retained having a much larger (hyper)volume than one would desire? This would drive down sample efficiency, especially as regions with posterior mass are driven towards the edge of this volume. This will _dramatically_ reduce sample efficiency in high dimensions, in a way that more sophisticated sequential methods are not affected. Furthermore, if the axis were rotated through 45 degrees (in all dimensions), then the volume of the hyper rectangle increases by a factor of 2^5=32, and hence the truncation will be a factor of 32x less efficient. This seems to be a major drawback to me.
> >
> > The truncation shape is tied to how TMNRE partitions the parameters into marginals, namely our choice of truncating with 1d marginals leads to hyperrectangles. There is flexibility in this decision, see reply to Reviewer 6wmD. For example, if we knew that certain parameters were likely to be highly correlated, then we could make sure those parameters get truncated together in a (2d+) marginal which considers those correlations. In this setting, it would be natural to use a more sophisticated truncation scheme for reasonable efficiency. An option for high dimension, currently under development, is application of overlapping hyperellipses like in [4].
> >
> > There are two major cases to consider: 1) When parameters are highly correlated and 2) When the posterior contains several modes. As you note (1) is inefficient with 1d truncation because the posterior mass is concentrated along the diagonal of a hyperrectangle, whereas a sequential method naturally incorporates this correlation into their sampling scheme and will shine on this example. The eggbox problem illustrates (2), without strong parameter correlations, that marginal and truncated treatment of many isolated modes is more efficient than SNRE.
> >
> > To address these concerns, we are creating a rotated version of the eggbox example inspired by your suggestion. We will include it in the supplemental materials. The correlations are still relatively minor but it will show the effects of rotation on TMNRE. We expect to have it done by Thursday as well but since it requires a new simulator, at the latest by next Monday.
> >
> > Our reply continues in the next point...

---

> > > ### Author Response · Authors · 2021-08-10
> > > **Major comments 4-8**
> > >
> > > > 4 - It seems to me that truncation may be a step back from the SNL-esque methods. I would prefer to see something along the lines of {SMC, MCMC}-MNRE, where instead of truncating the prior, a discrete/atomic representation is used. This would allow computation to be concentrated more efficiently, would relax the dependence on axis-aligned truncations, and prevent regions of parameter space from being aggressively deleted early in training. I realise this is not the paper that has been written, but it is a sticking point in my head that a set of strictly contracting axis-aligned truncations is just about the least flexible approach that could have been taken. You allude to this in the discussion (Lines 349-352). Do you have any further comments on this? This seems like the biggest weakness of the method. (I suppose you want to write Canonical TMNRE next…!)
> > >
> > > It all comes back to local amortization and our consistency check. Sequential methods are certainly efficient, especially in the case of highly correlated parameters; however, they do not offer our empirical consistency check, nor alternative quality checks. Meanwhile, amortized methods offer our empirical consistency check (and in the case of NRE, an additional receiver operating curve diagonistic [5]) but are rather inefficient. The purpose of our method is to find something in the middle, i.e. more efficient than fully amortized but more testable than sequential. It is indeed a tradeoff but we think it to be a worthwhile one for common usecases, namely expensive simulators.
> > >
> > > [5] Hermans, et. al., Likelihood-free MCMC with Amortized Approximate Ratio Estimators
> > >
> > > > 5 - I do not understand these “consistency checks through local amortization”. I figure you are suggesting that through the use of synthetic data we can test the inference result by recovering the ground truth. If this is the case, this seems trivial/generally true to me and does not seem to be a real plus-point of your method in particular. The real issues with methods such as this arise from model-mismatch in the generative model, and hence there are no real tests for this. I think this claim is overstated, and unless I’ve gotten something wrong, I would rather see this removed. (It could be referenced in passing as a sanity check or for debugging -- but nothing more). I could not follow Section 3.3 as a result.
> > >
> > > We're glad you brought up this important point. Just like other simulation-based inference methods, the problem of model mismatch is a legitimate concern. We implicitly assume that we are modeling the generative process correctly and we don't check if it is consistent with real data. Our empirical check accesses if the posterior is similar as if we would do the analysis with other synthetic data. To clarify this concern we added a sentence after line 294, `This empirical test is designed to show the consistency of the estimated nominal credible intervals across realizations of fabricated data but it does not address a generative model mismatch or access whether the nominal posterior corresponds to the ground truth posterior.`.
> > >
> > > That being said, we do not have the ground truth posterior and we cannot recover it with synthetic data. Rather, we intend to use (local) amortization to check the empirical versus nominial credible intervals predicted with out method. Let the independent variable be the nominal credibility contour and simulate parameter-simulation pairs. For every pair, determine whether the parameter lies above the nominal credibility contour or below it. The fraction which lied above the contour is the empirical credibility. A positive result is when the empirical and nominal contours agree, but it does not necessarily mean that estimated posterior corresponds with the ground truth. For example, choose the 68% highest posterior density region. A passing method puts 68% of the parameters from the parameter-simulation pairs above this contour. This calculation is natural for an amortized techinique but extremely expensive with a sequential technique.
> > >
> > > Local amortization comes with a caveat, namely, simulations from parameters at the edge of our truncated region may have posterior mass inaccurately truncated. We consider this part of the tradeoff from the increased efficiency that truncation offers. To make certain this point was made, we added a reference to the limitations section after line 294 and a paragraph discussing this caveat, see reviewer 6wmD main points, first bullet.
> > >
> > > We hope that this addresses your concern that these consistency check are generally available / applicable and rather that they feature as a positive aspect of our method, not shared by sequential ones.
> > >
> > > > 6 - According to Algorithm 1, the data points aren’t drawn from the prior, they are drawn from the truncated prior, and so is the ratio estimator targeting the correct quantity? This may just be a typo, i.e. Line 5 and Line 7 are identical (modulo a dash), whereas I believe Line 7 should drop the indicator term.
> > >
> > > The lines 5 and 7 correctly show how our algorithm works. The idea here is to draw both parameter samples from the same truncated prior distribution. The primed parameters are independently drawn from the data.
> > >
> > > > 7 - Using the C2ST score seems a little biased to me. Since the other methods are computing full posteriors, and yours is only computing a single marginal, performing a two-sample test comparing yourself to the full posterior actually seems a little harsh on your own method -- even _incorrect_ as a metric! I suppose this shows your method is working admirably, but also shows that maybe the solutions just happen to admit a factorized posterior (and hence are amenable to your method). Either way, I find using this as the only performance metric a bit of a head-scratcher. As mentioned above, I would like to see a more diverse suite of performance metrics examined.
> > >
> > > To clarify this, we turn our attention to our reply to question 1. The C2ST-ddm scores are computed as a mean across each d dimensional marginal since we don't estimate the joint. In the end the C2ST is computed only across one- and two-dimensional distributions.
> > >
> > > Among the tests in Figure 1, only the Gaussian Linear Uniform task admits a factorizable posterior. TMNRE estimates the marginals of a posterior with arbitrary correlations. We consolidated our discussion about other metrics above in our reply to Summary+. For a bit more, see lines 724-732.
> > >
> > > > 8 - You recover N^2 marginal pairs, and N marginals. The N^2 + N marginals that you recover will all be dramatically different as a result of marginalizing over N-2 other variables. Practically speaking, how do you select which distribution to draw from for a particular distribution? Moreover, how do you actually go about generating a sample from the joint...?
> > >
> > > In Algorithm 1, we find the truncated region with the N one-dimensional marginals. At that point, we produce the two-dimensional marginals in order to create a corner plot; however, any arbitrary marginal could be estimated directly from that the data within the truncated region. Practically, one selects the marginal posterior of interest to draw samples from a particular set of parameters. With this truncation method there are likely to be redundant estimates for a parameter, but unique estimates for collections (marginals) of parameters.
> > >
> > > As we mentioned in our reply to question 3, it is also possible to truncate with higher dimensional marginals but we did not write about it in this paper. In the paper we never sample from the joint, although it would be possible if one estimated the joint on the truncated region.

---

> > > > ### Author Response · Authors · 2021-08-10
> > > > **Other comments +**
> > > >
> > > > ### Other Comments
> > > > > A - Are the authors able to comment on the difference between their method and that of Jeffrey & Wandelt? I believe that the work is sufficiently different to be easily classified as novel throughout, but I think a more substantiated explanation of any similarities and differences would be useful.
> > > >
> > > > The paper by Jeffrey & Wandelt proposes two algorithms: 1) moment networks which estimate the moments of a distribution to find (for example) the best Gaussian approximation to the posterior and 2) a normalizing flow which estimates the marginal posterior directly. Its closest relative would be a marginal version of Neural Posterior Estimation, but they provide no reference to sequential techniques nor truncation.
> > > >
> > > > > B - Does truncating the prior lead to sections of parameter space being permanently deleted? I don’t think the algorithm could recover if a section with mass were not covered. I realise this is true for most sampling algorithms, and Fig 4 suggests that the distribution starts broad and then narrows. But it strikes me as though it could be particularly dangerous in this setting if there is categorically no way to expand the distribution. Maybe consider adding some discussion of entropy regularization to the prior to the conclusion.
> > > >
> > > > This is a very valid concern and we have made choices (like using NRE and a very small epsilon) to be conservative with truncation for exactly this reason, see our reply to major point 1. We haven't observed this failure mode and it can be avoided by sampling more at earlier stages. Our posterior equicontour check (lines 823 and 824) can help determine if mass has been left out. (That check doesn't work if an entire mode is omitted.) We made sure to include this caveat in line 823, `The caveat is that if an entire separate mode is truncated, this test will not indicate it.`. In general, this is less of an issue in lower dimensions because "losing" mass in the hypervolume is less likely.
> > > >
> > > > > C - There is a relatively substantial limitations section in the supplementary materials. I would prefer it if some of these limitations were brought up and explored in the main text.
> > > >
> > > > This is a reasonable request and agreed upon across reviewers. For that reason, we have included it our official statement as a priority to update the discussion section with a more thorough limitations section in the camera ready version of the paper.
> > > >
> > > > > D - The cosmology experiment is by far the most interesting of the experiments. I do not understand why it was demoted to the supplementary materials.
> > > >
> > > > We thank the referee for the positive words about the cosmology example. We addressed it above in the Summary section, but we restate here that we will prioritize clarifying amortization, explaining caveats, and finally the cosmology example, given page space.
> > > >
> > > > > E - There also seems to be a missed opportunity using the marginals recovered to establish independencies or conditional independencies in the posterior, where this method explicitly enforces the efficiency gains that these independences can provide.
> > > >
> > > > The reviewer notes an interesting point that perhaps this tool could be used to determine independencies within the posterior itself. As written, TMNRE works to provide marginal posterior estimates over a joint which can have arbitrary independence and conditional independence properties.
> > > >
> > > > > F - An interesting point that is not strongly emphasised is that the same samples can be (re-)used to learn all of the ratio estimators.
> > > >
> > > > The authors wholeheartedly agree that this represents one of the key features of our method. That way the user can truncated in one-dimension and then reuse the data to estimate 2d+ marginal likelihood-to-evidence ratios. You are quite right and it was not emphasized enough in the text. On line 171 we added the sentence, `We emphasize that the data already generated during the truncation phase can be easily reused to learn arbitrary marginals of interest. When higher accuracy likelihood-to-evidence ratio estimates are called for, the user can simulate from within the truncated region.`.
> > > >
> > > > > G - I am also a little confused about the “local amortization”. This method is inherently sequential and conditioned on a single xo, and so there is no real amortization going on. Around Line 190, you say that you train “locally amortized posteriors that are valid for parameters $\theta \in \Gamma(m)$”. Are suggesting that the function approximators are going to be someway accurate/informative over the entire domain of $\Gamma(m)$? This should be explained more clearly if this is the case.
> > > >
> > > > We consolidated our reply to this question within point 5 above and hope that this was fully addressed there.
> > > >
> > > > ### Minor / Typographical Comments
> > > >
> > > > > - Inline or wrapped figures should be avoided. They are bad for readability and flow. Figures should really be floated to the top (or bottom at a push) of the page.
> > > >
> > > > The reviewer makes a good point about readability. If we have space we will remove wrapped figure.
> > > >
> > > > > - I don’t understand the sentence beginning Line 30: “With existing…”.
> > > >
> > > > This is an important sentence to clarify. We have changed it to explicitly reference increased accuracy per simulation (sequential methods) and empirically testable results (amortized estimates, non-sequential methods) in the camera ready version. For more info, see our reply to point 4.
> > > >
> > > > > - Line 129: I don’t understand how a likelihood ratio estimator can be overconfident...!
> > > >
> > > > This is indeed confusing, so we made sure to clarify this in the camera ready text where we added the explanation `/ narrower than the ground truth` to the parenthetical.
> > > >
> > > > > - Line 147: Incorrectly rendered characters (I only see boxes).
> > > >
> > > > This is a bit of an unusual notation, so we made sure to adjust the text to clarify that the boxes intentionally indicate the effects of the subscript $\Gamma$ on an arbitrary symbol.
> > > >
> > > > > - Some of the font sizes in the figures are TINY.
> > > >
> > > > This is another good point regarding readability. We will emphasize larger caption size in the camera ready version!
> > > >
> > > > > - I don’t really understand what Figures 4(b-d) are showing.
> > > >
> > > > The figures are a visual representation of our so-called empirical consistency check over different highest posterior density intervals (see above, point 5). For a given nominal credible interval (x-axis), the y-axis reports the empirical credibility, as measured by fraction of case that the true parameter lies within that nominal credible interval. For us, it is desirable that the line is at or above the diagonal because that implies conservative nominal credible intervals. Hopefully this is further clarified by our reply to point 5. We made sure to change the figure caption to include `The line at (above) the diagonal implies accurate (conservative or wide) nominal credible intervals. See section \ref{sec:empirical_tests}.`.
> > > >
> > > > > - Figure 6 does a much better job of showing the truncation compared to Figure 4a.
> > > >
> > > > The authors agree with the reviewer, although Figure 4a does not intend to show truncation alone, rather it shows the evolution of the posterior estimate between rounds. Since the figure is unclear, we have changed the caption accordingly to emphasize its purpose. `The ratio estimator for $\theta_2$ in the 3-dim torus example, for several consecutive rounds. The estimator is conservative with limited data because each round's estimate converges from above to the final results (red line); i.e. the early rounds produce too-wide posteriors thus truncation decisions are made safely.` While Figure 6 is clearer about truncation, it does not show the evolution of the estimate across rounds.
> > > >
> > > > > - The discussion is very long and doesn’t add much (partly because the rest of the paper is pretty well written). This could be cut down and used for the cosmology example instead, or, for Figure 6.
> > > >
> > > > We agree that the discussion can be trimmed. We will reduce its size and take the space to include the changes above as well as focus clarification about amortization with the remaining space and on algorithmic caveats. As mentioned in point D, we will consider including the cosmology example if there is space.
> > > >
> > > > > - The societal impact statement is too long. I think it is safe to say here: “There are no societal impacts.” (as is explicitly permitted in the submission instructions).
> > > >
> > > > In general, the authors agree with the reviewer that societal impacts are marginal. We will aim to reduce the size of our response to make space for other important additions, like the cosmology example.
> > > >
> > > > ## Limitations And Societal Impact
> > > > > No societal impacts of note.
> > > >
> > > > ## Closing Statement
> > > >
> > > > Thank you Reviewer BG8g for the positive score, constructive comments, and important questions. Your review included a range of topics including points in regards to our empirical check, the choice of truncation strategy and its relationship to sequential methods, and fundamental aspects of simulation-based inference like model mismatch, among other topics. We will include the experiments like the rotated eggbox and baseline performance on the eggbox and torus models. The changes you suggested are reflected in our camera ready document and politely ask that you consider increasing your score as suggested in the "Summary+" section. If there any follow up questions or changes, please let us know.

---

> ### Author Response · Authors · 2021-08-12
> **New experimental results**
>
> The eggbox experimental results were already given within our initial reply. NLE and SNLE performed well, while NPE did not resolve the modes. After over 48 hours, SNPE has failed to produce the necessary samples on the eggbox task (still running), likely indicating an inaccurate posterior.
>
> The torus posteriors have finished. The results were in line with our predictions. Non-sequential methods all failed to accurately represent the posterior. NPE and NRE produced a mass in the correct location but distorted and no "doughnut hole." NLE placed posterior mass completely in the wrong part of the parameter space. The SNLE and SNRE produced accurate posteriors. SNPE accurately represented the high density region but the tails have some distortion.
>
> (Also replied to opr4)
>
> The rotated eggbox experiment is not yet complete. We will update you when it is done next week.

---

> > ### Author Response · Authors · 2021-08-16
> > **rotated eggbox**
> >
> > The rotated eggbox experiment is still in the works and we will provide you with results as soon as it is done. In the mean time, we encourage you to read our text response to this matter where we discuss the principles relevant to non-axis aligned posteriors before we provide experimental data. (Furthermore, there are non-axis aligned posteriors within the sbi benchmark.)

---

> > > ### Author Response · Authors · 2021-08-19
> > > **rotated eggbox results**
> > >
> > > We created a rotated version of the eggbox simulator where the initial simulator $g(\theta)$ was replaced by $g(Q^{T} \theta)$. $Q \in \mathbb{R}^{10 \times 10}$ is a rotation matrix which rotates the point $(1, 1, 1, ..., 1)^{T}$ to $(0, 0, ..., c)$ where $c$ is a positive constant. We had been using a uniform prior defined by the unit cube; however, this region no longer holds the posterior mass. We determined the limits of a hyperrectangle which completely covered the rotated prior in the following manner: Consider each unit vector pointing to the edge of the unit cube $E = \times_{n=0}^{10} \{0, 1\}$ where $\times$ denotes the Cartesian product. If each of those vectors $v_i$ is rotated by $Q$, $v_i \mapsto Qv_i$, we can then look at the minimum and maximum values projected along each basis vector and set those to be the limits of our new uniform prior. The bounds were $[(-0.924, 0.924), ..., (-0.924, 0.924), (0.000, 3.16)]]$ which implies a volume 795 times larger than the initial prior. Rotation of high dimensional objects can produce unexpected results when projected down to 2d and 1d. The ground truth posterior samples had four distinct "blobs" of probability mass in 9 out of 10 dimensions, but they were sheared and bled into each other. Furthermore, each "blob" of probability had multiple submodes within it. The final dimension produced two regions of high probability each streaked with many modes. When projected in a corner plot, the posterior did not look axis-aligned along any dimension.
> > >
> > > It's important to remember that this posterior is periodic. That implies our method will not just find the solutions within the rotated hypercube, but also any other solutions in the larger axis-aligned cube which covers the entire rotated hypercube. For that reason, we also produced translated versions of the ground truth posterior samples then rotated those. Most 2d projections looked like diagonal streaks of density. The final row in the corner plot had a longer period and was orthogonal to the other projections' density and contained patches of zero density.
> > >
> > > Just like the initial formulation of the eggbox, we provided 100,000 training samples. NRE, SNRE, and MNRE all failed to converge. The posteriors looked like noise without structure for all but SNRE. SNRE found the a noisy version of the structure in some 2d projections but produced noise in most. Since NLE and SNLE were effective on the non-rotated eggbox posterior, we also trained NLE and SNLE using the same number of training samples. NLE did not converge. SNLE approximated the ground truth streaks in 2d better than any other method but still had regions of missing probability mass. The final row in the corner plot did not have the correct structure.
> > >
> > > In other words, the rotated version of this problem is too difficult to solve at this budget. This took longer than expected because the sequential methods required over 48 hours of computation.

---

> > > > ### Comment · Reviewer_BG8g · 2021-08-31
> > > > **Response**
> > > >
> > > > Thank you to the authors for their lengthy and detailed response, and for getting on it and running so many new experiments in such a short time.  It sounds like the results are broadly positive, but without being able to see the results, settings etc I am hesitant to upgrade my score based solely off of this discourse.  I also still have some reservations about the true utility and generality of the method, and how much it really contributes to LFI, but, I am certainly not anti acceptance.  Therefore I am happy if the paper were to be accepted, but I will not ardently fight for its acceptance.  Good luck.

---

### Official Review · Reviewer_opr4 · 2021-07-15

**Rating:** 7
**Confidence:** 4

**Summary:**

This study contributes with a new simulation-based inference method, Truncated Marginal Neural Ratio Estimation (TMNRE). It builds on a previous method, Neural Ratio Estimation (NRE), by combining NRE (for directly inferring marginals of the posterior distribution) and the sequential constraining of the prior distribution by truncation (targeted to a particular observation). On standard tasks, TMNRE is shown to perform on par with state-of-the-art methods. Furthermore, the sequential truncation of the prior allows for local amortisation and thus makes it easy to perform (local) consistency tests.

**Limitations And Societal Impact:**

The authors have for the most part adequately addressed the limitations of their work. However, I would like to raise a point regarding the limitations of estimating marginals rather than the full posterior: arguably, in many cases the practitioner of simulation-based inference would indeed be interested in having good posterior predictives. These posterior predictives are not possible with TMNRE (as the authors acknowledge in line 345), and I would recommend the authors to more explicitly acknowledge this potentially impactful limitation.

**Main Review:**

### Originality

The novelty of the method consists in the sequential constraining of the prior distribution by truncation. Previous work is for the most part appropriately cited and discussed, and it is clear how this work differs from previous contributions. However, I believe it would be important to also discuss the paper by Blum and François (2010 - Non-linear regression models for Approximate Bayesian Computation), as this proposes an ABC method also with prior truncation.


### Quality

The paper is technically sound, and claims are mostly well supported by empirical evaluation. However, I have a couple of comments I would appreciate the authors to address:

-the authors write "Targeting marginals directly, by estimating only the marginal posterior for the parameters of interest, is simpler and sufficient for most purposes." As a first minor point, I believe "most purposes" to be quite an overstatement, and would recommend the authors to instead write "many" or something similar (see also a related point under section "Limitations");

-throughout the manuscript, the authors put quite some emphasis on the fact that TMNRE is more efficient, given its sequential prior truncation feature. However, while this efficiency is convincingly shown against MNRE (no truncation), NRE and some other variants (Figures 2,3,5), it would have been important to also show this against other methods (e.g. SNLE, SNPE), and for several simulation budgets.


### Clarity

The paper is clearly written, and provides enough information for an expert reader to understand all the steps to reproduce the results. I would just like to point out a few typos:

-lines 161-162, rather than "r(x|theta)", "r(x,theta)";

-Figure 2, right panel, the x-axis is in units of the common logarithm, and that should be specified;

-the authors write "That means simulations from within this constrained region are likely informative while simulations from outside the region are likely uninformative". I would suggest the authors to rephrase this sentence, as the wording "(un)informative", without stating about what, is a bit imprecise and potentially misleading.



### Significance

This new method will be of interest to the simulation-based inference community and inspire further method development.


### After rebuttal

Thank you for the careful answer to my review. After reading all the reviews and authors' rebuttal, I believe that this is an interesting paper and that the authors made an effort to address all the reviews, and in my opinion, to satisfaction. While TMNRE is a method for the exploration of the posterior (or more concretely, its marginals) rather than a method for full Bayesian inference, I think this will be a valuable/useful addition to the SBI toolkit. I increased my score accordingly.

**Time Spent Reviewing:**

3

---

> ### Author Response · Authors · 2021-08-10
> **Initial comment - opr4**
>
>
> ## Summary
>
> > This study contributes with a new simulation-based inference method, ... (TMNRE). It builds on a previous method, Neural Ratio Estimation (NRE), by combining NRE (for directly inferring marginals of the posterior distribution) and the sequential constraining of the prior distribution by truncation (targeted to a particular observation). On standard tasks, TMNRE is shown to perform on par with state-of-the-art methods. Furthermore, the sequential truncation of the prior allows for local amortisation and thus makes it easy to perform (local) consistency tests.
>
> The authors offer their gratitude to Reviewer opr4 for reading and distilling the essence of our paper, the positive assessment, and the helpful comments.
>
> ## Main Review
> ### Originality
> > The novelty of the method consists in the sequential constraining of the prior distribution by truncation. Previous work is for the most part appropriately cited and discussed, and it is clear how this work differs from previous contributions. However, I believe it would be important to also discuss the paper by Blum and François (2010 - Non-linear regression models for Approximate Bayesian Computation), as this proposes an ABC method also with prior truncation.
>
> The reviewer is correct to note that the work by `Blum and François (2010)` is relevant due to their use of a truncated prior distribution as introduced in that paper's "Iterated importance sampling" section. Therefore, we added a statement to summarize this paper in our related work section about ABC itself on line 67. The new statement reads, `Blum and Fran\c{c}ois \cite{blum2010non} introduce an \ABC\ distance criterion weighting mechanism to tune the posterior sampler as well as a proposal prior which draws from a truncated region of true prior. It estimates the support of previously-accepted samples via support vector machines \cite{scholkopf2002learning} and samples from this region with rejection.`
>
> ### Quality
>
> > The paper is technically sound, and claims are mostly well supported by empirical evaluation. However, I have a couple of comments I would appreciate the authors to address:
> >
> > - the authors write "Targeting marginals directly, by estimating only the marginal posterior for the parameters of interest, is simpler and sufficient for most purposes." As a first minor point, I believe "most purposes" to be quite an overstatement, and would recommend the authors to instead write "many" or something similar (see also a related point under section "Limitations");
>
> The authors have changed the relevant paragraph in line 37 to be more specific about the use case and therefore blunt the overstatement. The paragraph in question now reads, `Scientific insight is often based on a low dimensional marginalization of the posterior with nuisance parameters removed \cite{aghanim2020planck}. The additional information of the full joint posterior might not be worth the additional cost afforded. Targeting marginals directly, by estimating only the marginal posterior for the parameters of interest, is simpler and sufficient for many scientific, parameter estimation, and bounding purposes.`
>
> where `\cite{aghanim2020planck}` means Aghanim, Nabila, et al. "Planck 2018 results-VI. Cosmological parameters." _Astronomy & Astrophysics_ 641 (2020): A6.
>
> > - throughout the manuscript, the authors put quite some emphasis on the fact that TMNRE is more efficient, given its sequential prior truncation feature. However, while this efficiency is convincingly shown against MNRE (no truncation), NRE and some other variants (Figures 2,3,5), it would have been important to also show this against other methods (e.g. SNLE, SNPE), and for several simulation budgets.
>
> In the present manuscript, we focused on extensions of NRE because we found empirically that NRE typically overestimates the support of the posterior in the presence of limited training data, which is important to not prematurely remove regions of the parameter space when using the technique in multiple rounds to truncate the posterior.  We did not find the same consistent behavior for NLE or NPE (although we did not perform a systematic study for these algorithms, and hence did not emphasize this in our manuscript).
>
> Reviewer BG8g mentioned the same point, suggesting to apply other method to our eggbox and torus problems. These are currently underway, the results will be added to the camera-ready version, and we will post our findings here on Thursday. To speculate a bit...
>
> On the torus problem, given our fixed simulation budget, the authors predict that sequential methods will perform well while non-sequential methods will not receive the simulations necessary to accurately represent the posterior, as seen with MNRE in Figure 2.
>
> The eggbox experiment with SNLE, NLE and NPE is already complete (still waiting on SNPE). NPE fails to accurately model the eggbox at the simulation budget, meanwhile SNLE and NLE model the posterior very accurately. This result will be included along with an explanation in a supplemental section with a reference in the main text. The text will be provided here when all experiments are complete.
>
> To briefly reply... NPE cannot handle this joint distribution and would perhaps be better treated marginally, like we do with TMNRE. NLE and SNLE's performance indicates the effectiveness of simulation-based likelihood estimation, even on tough problems like the eggbox. Despite the strong performance, likelihood estimation requires fitting a flow to the data distribution which is difficult when the data is complicated, as it is by scientific practitioners who are our intended audience. An example would be the data from the cosmology simulator. Furthermore this effect has been documented in the SLCP Distractors example in [1] where likelihood estimation was confounded by extraneous information in the data.
>
> ### Clarity
> > The paper is clearly written, and provides enough information for an expert reader to understand all the steps to reproduce the results. I would just like to point out a few typos:
> > - lines 161-162, rather than "r(x|theta)", "r(x,theta)";
>
> Good find! We propose to use an alternative. We find writing the likelihood-to-evidence ratio is clearer as  $r(x \mid theta)$. Therefore, we changed the introduction of the symbol in (2) and line 127 to conform to $r(x \mid theta)$. All instances have been renamed in this fashion for the camera ready version.
>
> > - Figure 2, right panel, the x-axis is in units of the common logarithm, and that should be specified;
>
> The reviewer is correct and the fourth panel in figure 2 has been updated to read $\log_{10} \epsilon$.
>
> > - the authors write "That means simulations from within this constrained region are likely informative while simulations from outside the region are likely uninformative". I would suggest the authors to rephrase this sentence, as the wording "(un)informative", without stating about what, is a bit imprecise and potentially misleading.
>
> We agree and have changed the sentence on lines 327 and 328 to `That implies parameters from within this constrained region are more likely to produce data similar to $x_{o}$ than simulations from outside the region.`.
>
>
> ### Significance
> > This new method will be of interest to the simulation-based inference community and inspire further method development.
>
> ## Limitations And Societal Impact
> > The authors have for the most part adequately addressed the limitations of their work. However, I would like to raise a point regarding the limitations of estimating marginals rather than the full posterior: arguably, in many cases the practitioner of simulation-based inference would indeed be interested in having good posterior predictives. These posterior predictives are not possible with TMNRE (as the authors acknowledge in line 345), and I would recommend the authors to more explicitly acknowledge this potentially impactful limitation.
>
> In an effort to more explicitly acknowledge this important limitation of TMNRE we have added a new paragraph in the TMNRE method section between lines 182 and 183 which reads as follows, `Algorithm~\ref{alg:tmnre} estimates the necessary marginal posteriors for corner plot visualization. We demonstrate the cost-effectiveness of this algorithm in Section~\ref{sec:experiments}. An important limitation of Algorithm~\ref{alg:tmnre} is the inaccessibility of the posterior predictive distribution. This limitation is mitigated by training a ratio estimator on all parameters within the truncated region; however, producing an accurate joint estimate may come with (sometimes significant) additional simulation costs.`
>
> We also note that this particular limitation was not represented in the limitation section of our Appendix, despite including it in the discussion. We have now included it in the limitations section. It reads as follows, `Finally, as mentioned in Section~\ref{sec:tmnre}. Algorithm~\ref{alg:tmnre} does not naively allow for sampling from the posterior predictive distribution $p(\bx' \mid \bx) \coloneqq \int_\Omega p(\bx' \mid \btheta) p(\btheta \mid \bx) \, d\btheta$ because it only produces the one- and two-dimensional marginals. It remains possible to estimate the joint posterior within the truncated region by simply training another ratio estimator with all parameters. Doing this, using only the simulations necessary for Algorithm~\ref{alg:tmnre}, may produce an inaccurate joint posterior estimate.`
>
> ## Closing Statement
>
> We thank Reviewer opr4 for their thoughtful peer review and positive rating. The reviewer noted out the connection to the `Blum and François (2010)` paper, clarified the set of circumstances in which our method would be applicable, and suggested finding the baseline results on our new examples. Hopefully our response has clarified that your concerns are addressed by stated changes in the camera ready version. If you have anything else you'd like to discuss, please let us know!

---

> ### Author Response · Authors · 2021-08-12
> **New experimental results**
>
> The eggbox experimental results were already given within our initial reply. NLE and SNLE performed well, while NPE did not resolve the modes. After over 48 hours, SNPE has failed to produce the necessary samples on the eggbox task (still running), likely indicating an inaccurate posterior.
>
> The torus posteriors have finished. The results were in line with our predictions. Non-sequential methods all failed to accurately represent the posterior. NPE and NRE produced a mass in the correct location but distorted and no "doughnut hole." NLE placed posterior mass completely in the wrong part of the parameter space. The SNLE and SNRE produced accurate posteriors. SNPE accurately represented the high density region but the tails have some distortion.
>
> (Also replied to BG8g)

---

### Official Review · Reviewer_6wmD · 2021-07-28

**Rating:** 7
**Confidence:** 4

**Summary:**

This work proposes a novel algorithm for estimating the posterior distribution in simulation-based inference (SBI) settings, where the likelihood is not known but it is implicitly defined via a stochastic simulator. The work proposes truncated marginal neural ratio estimation (TMNRE), which is able to approximate the posterior distribution only for a small number of parameters of interest and making the sampling more efficient by sequentially truncating the prior for a given observed data. In addition, the posterior distribution is amortized and can be used to check the goodness of fit of the posterior distribution, albeit only in the areas which have not been truncated during training. The paper offers extensive experiments for different parameter dimensionality, as well as a comparison with the state of the art in SBI and theoretical analysis of how much error is introduced with the prior truncating process.

**Limitations And Societal Impact:**

Yes, limitations and societal impact have been addressed adequately.

**Main Review:**

I found the paper clear and easy to read, with the main algorithm being presented clearly (the comments in the algorithm box are much appreciated), as well as the relationship with related work. The experimentation is extensive and tackles different settings that could be challenging: an asymmetric low-dimensional distribution and a high-dimensional multimodal example, in addition to a series of standard tasks in SBI which have been recently established by [1] and a physics-motivated example in Appendix E.
Additionally, the authors elaborate on how much the \epsilon impacts the approximation error in a couple of settings, which helps the reader get an idea of the tradeoff in posterior inference; having said that, the fact that the \epsilon is set to be very low likes make a negligible difference in inference (at least in the example shown in the paper).

Overall, I feel this is a solid paper and would be a great addition to the SBI community.
I am including some (minor) comments below:

- The amortization for every data and parameter value is a nice property, but it seems like it is restricted to the constrained region of the prior. Given the truncation in this paper seems to be pretty mild (in terms of tail trimming), I assume that checking the posterior fit for different observed data would still not be affected by the truncation. Can the author comment on how empirical check would be affected by a more aggressive trimming?

- The truncation idea is quite interesting and potentially beneficial to other methods. As the author observed, creating a hyper-rectangle is not the most efficient way of doing truncation, but I assume it's probably computationally faster as one does not need to consider the correlation between parameters. Would it be possible for the truncation mechanism to just consider the correlation between the parameters of interest and doing 1D truncation among the nuisance parameters as a tradeoff? In addition, can the authors comment on how many initial samples one should use in order to avoid missing a potentially relevant area of the parameter space and how much is this number affected by the dimensionality of the parameter space?


[1] Lueckmann et. al, "Benchmarking Simulation-Based Inference", AISTATS 2021



**Time Spent Reviewing:**

3.5

---

> ### Author Response · Authors · 2021-08-10
> **Initial comment - 6wmD**
>
> # Official Comment - 6wmD
> ## Summary
> > This work proposes a novel algorithm for estimating the posterior distribution in simulation-based inference (SBI) settings, where the likelihood is not known but it is implicitly defined via a stochastic simulator. The work proposes truncated marginal neural ratio estimation (TMNRE), which is able to approximate the posterior distribution only for a small number of parameters of interest and making the sampling more efficient by sequentially truncating the prior for a given observed data. In addition, the posterior distribution is amortized and can be used to check the goodness of fit of the posterior distribution, albeit only in the areas which have not been truncated during training. The paper offers extensive experiments for different parameter dimensionality, as well as a comparison with the state of the art in SBI and theoretical analysis of how much error is introduced with the prior truncating process.
>
> We are highly appreciative of the time Reviewer 6wmD donated to review our paper and are very thankful for the thoughtful comments and positive assessment.
>
> ## Main Review
> > I found the paper clear and easy to read, with the main algorithm being presented clearly (the comments in the algorithm box are much appreciated), as well as the relationship with related work. The experimentation is extensive and tackles different settings that could be challenging: an asymmetric low-dimensional distribution and a high-dimensional multimodal example, in addition to a series of standard tasks in SBI which have been recently established by [1] and a physics-motivated example in Appendix E. Additionally, the authors elaborate on how much the \epsilon impacts the approximation error in a couple of settings, which helps the reader get an idea of the tradeoff in posterior inference; having said that, the fact that the \epsilon is set to be very low likes make a negligible difference in inference (at least in the example shown in the paper).
> >
> > Overall, I feel this is a solid paper and would be a great addition to the SBI community. I am including some (minor) comments below:
> >
> >  - The amortization for every data and parameter value is a nice property, but it seems like it is restricted to the constrained region of the prior. Given the truncation in this paper seems to be pretty mild (in terms of tail trimming), I assume that checking the posterior fit for different observed data would still not be affected by the truncation. Can the author comment on how empirical check would be affected by a more aggressive trimming?
>
> Only data generated by parameters that have posterior mass within the constrained region can be accurately represented by a trained ratio estimator. A general goal when applying our algorithm would be to keep truncation wide enough to not affect empirical tests.  However, it is important to note that the test merely indicates whether or not there is agreement between the nominal and empirical credible intervals, but not necessarily agreement between the estimated distribution and the ground truth marginal posterior. For this reason, the test may still indicate empirical and nominal agreement even when epsilon is too large (and trimming too aggressive). We offer another solution to note a too-large epsilon (see lines 823 and 824).
>
> We added some text to clarify this after line 294, `When $\epsilon$ is small enough, the effects of truncation will not significantly impact this empirical test because of the accurate posterior estimate. Large $\epsilon$, that trim the tails of the estimated posterior aggressively, render this test unreliable because the estimated likelihood-to-evidence ratios will have inaccurate highest density intervals. It is possible to check whether $\epsilon$ is too large as indicated by a high-density posterior equicontour line intersecting with the truncation bounds, see Appendix~\ref{sec:appendix_limitations}.`
>
> To further increase clarity about the effects of aggressive trimming in general, we added the following text after line 820, `To illustrate this point, consider a simulator with a two-dimensional parameter space. If significant amounts of posterior mass are truncated in the $\theta_0$ dimension, the ground truth $\theta_1$-marginal posterior, under the truncated prior, transforms from the intended marginal distribution $p(\theta_1 \mid x)$ towards a conditional distribution $p(\theta_1 \mid x, \theta_0^\ast)$. Where $\theta_0^\ast$ denotes the center of the truncated region in the $\theta_0$ dimension.`
>
> >  - The truncation idea is quite interesting and potentially beneficial to other methods. As the author observed, creating a hyper-rectangle is not the most efficient way of doing truncation, but I assume it's probably computationally faster as one does not need to consider the correlation between parameters.
>
> The authors agree that the truncation process is not uniquely suited to NRE, rather NLE or NPE might be able to benefit as well. The reviewer correctly notes that it is computationally simpler to use hyperrectangles, although other shapes might prove more effective (see lines 815-817). The authors note that using other truncation shapes is a project currently under investigation and will be covered in future work.
>
> This was a point that reviewer BG8g also made and we will make sure that these properties are emphasized in the discussion section. Our choices and the corresponding consequences will be explained.
>
> > Would it be possible for the truncation mechanism to just consider the correlation between the parameters of interest and doing 1D truncation among the nuisance parameters as a tradeoff?
>
> Yes, it is possible, and indeed one of the intended future use cases! The authors posit a few options: truncate in 1d with parameters and 1d with nuisance parameters, truncate in Nd with parameters and 1d with nuisance parameters, or truncate in Nd with parameters and Nd with nuisance parameters. In addition, autoregressive formulations may be an option. This has not been explored extensively, although we have performed some exploratory experiments with truncating in 2d using multiple overlapping circles as bounds. In contrast to hyperrectangles, a hyperellipse could greatly increase sampling efficiency of, for example, an extremely diagonally correlated Gaussian posterior. In this case, most of the hyperrectangle's volume would be in low density posterior regions, but not so for the hyperellipse.
>
> > In addition, can the authors comment on how many initial samples one should use in order to avoid missing a potentially relevant area of the parameter space and how much is this number affected by the dimensionality of the parameter space?
> >
> >  [1] Lueckmann et. al, "Benchmarking Simulation-Based Inference", AISTATS 2021
>
> This is an important question for practitioners to use the method effectively. On one hand, too many initial simulations leads to more computational waste while too few can lead to inaccurate posteriors. As a heuristic, (see appendix A.5) we applied approximately a third of the simulation budget in the initial sampling phase.
>
> In the scientific setting there is not usually a specific budget constraint, rather an emphasis on reducing computational waste to increase discovery rate. In this case, we would recommend simulating at the prior level until defined, but poorly resolved, regions of high density appear in the prediction.
>
> Regarding dimensionality. A sample drawn from a wide high-dimensional prior has a small chance of being within the H.P.D. region due to the curse of dimensionality. One major advantage of approximating marginals is that we are no longer dealing with high dimensional posteriors so many samples will lie within a one-dimensional marginal H.P.D. region of a subset of the parameters of interest.
>
> In likelihood-based settings one often encounters the situation that only very small regions of the parameter space have a high likelihood [Swendsen and Wang, Duane]. Likelihood-based algorithms (like Metropolis-Hastings or nested sampling techniques) must then first "find" these regions in order to explore them. A low number of initial samples (or e.g. "live-points" for nested sampling techniques) might mean that these regions are missed.  Unfortunately, this situation does not uniquely translate into a likelihood-free setting, since likelihoods are only implicitly defined through simulations. Different simulations can lead to the same effective likelihood function, and it depends on the details of the simulated data how hard it is to identify small high-likelihood regions.  In practice (e.g. the eggbox problem) we observed that the main failure mode of too few initial samples is typically not to miss relevant parameter regions, but instead for the network to be unable to train. In this case our algorithm returns simply the prior as posterior. However, in particular when the dimensionality of the simulated data is much lower than the dimensionality of the parameter space, the danger for missing a high-likelihood regions increases. In the worst case, we expect that the number of required initial samples will be similar to the number of required live-points when solving the same problem with a nested sampling algorithm.
>
> [Swendsen and Wang] Swendsen and Wang. (1986). Replica Monte Carlo simulation of spin glasses. Physical Review Letters 57 : 2607–2609.
> [Duane] Simon Duane, et. al. Hybrid Monte Carlo, Physics Letters B, Volume 195, Issue 2, 1987, Pages 216-222, ISSN 0370-2693, https://doi.org/10.1016/0370-2693(87)91197-X.
>
> ## Limitations And Societal Impact
> > Yes, limitations and societal impact have been addressed adequately.

---

> > ### Author Response · Authors · 2021-08-10
> > **Closing statement**
> >
> > ## Closing Statement
> >
> > We thank Reviewer 6wmD again for the positive review of our paper and for carefully considering our official statement. The reviewer brought up important points regarding the effects of aggressive truncation, our choices regarding truncation shape, and best practices to avoid truncating posterior mass. We hope our reply and changes made to the camera ready document served to alleviate any remaining issues. If there are other changes you would like to see, we would be glad to discuss them with you.

---

### Author Response · Authors · 2021-08-10
**Summary of reviews and response**

Thank you for the donation of time and expertise to all the reviewers. We will summarize the reviews, our responses and actions taken in the camera ready version, and conclude.

### Summary
1. Generally reviewers were satisfied with the presentation and coherence of the paper, especially the algorithm itself. Adjustments were suggested about some of the mathematical notation.
2. There were requests to clarify the usecase and limitations, i.e. we don't target the joint or posterior predictive. This is a limitation.
3. Requests for better explanation of the empirical check and its relationship with truncation / local amortization.
4. There was discussion about transferability of the method and comments about whether it was applicable to NPE and NLE, for example.
5. The truncation shape was of general interest.
6. Two reviewers thought baseline results on the eggbox and torus would be useful.
7. Two reviewers referenced the cosmology example in the appendix.
8. There were important reference suggestions.


### Actions Taken in Camera Ready Version
1. We made an effort to preserve coherence. Critiques about clarity were all incorporated, especially the ones about mathematical notation.
2. Initial claims about the usecase were clarified. Lack of joint / posterior predictive access were emphasized. The limitations section grew and references to it were made throughout the document. Further, we will update the discussion section to better reflect the usecase and caveats.
3. The section about the empirical check was expanded and the relevant caption was expanded too. Further, its usecase / limitations were clarified.
4. A small section was added about applicability to other methods in the introduction.
5. We clarified the effects of 1d truncation and how this leads to hyperrectangles. Limitations about this technique were explained in regards to highly correlated parameters. Future work with hyperellipses was noted.
6. We are currently running baselines on those examples. On the eggbox SNLE and NLE perform quite well while NPE fails, still waiting on SNPE. Expect remaining experimental results this Thursday. Discussion regarding this will be added to the supplemental materials. A critical point to support our choice of NRE, is the limitations of NLE on complex data, see SLCP Distractors in [1].
7. The cosmology example is instructive because it shows the intended usecase of TMNRE. It also requires a significant amount of space to explain. No matter what, we will make a much more visible reference to it in the camera ready version. If there is remaining space after clarifying the empirical consistency check and explaining the caveats of our method, we will include the cosmology example in the main text.
8. All suggested references were included and additional relevant ones were added to address recency bias. To clarify the usecase, we added cosmology citations where marginal posteriors are utilized to estimate physical parameters.

[1] Lueckmann et. al, "Benchmarking Simulation-Based Inference", AISTATS 2021

### Conclusion

We want to thank the reviewers for a generally positive assessment of our paper. Each point from your reviews were carefully considered and, hopefully, satisfactorily addressed along with with an appropriate change to the camera ready document. Considering these changes, we ask you to please revise your initial ratings in a fair and factual manner, taking our changes into account. If you would like to raise further concerns, we would be happy to respond to them!

---

### Decision · Program_Chairs · 2021-09-27

**Decision:**

Accept (Poster)

**Comment:**

The paper proposes a method for simulation-based inference that is based on neural ratio estimation and estimates marginals of the posterior distribution.

The reviewers are generally positive about the submission, citing the usefulness of the proposed method, the clarity of the presentation, and the good variety of experiments as strengths. There were some initial concerns, but the reviewers appreciated the extensive and thorough responses provided by the authors and most concerns were allayed. Overall, I'm happy to recommend acceptance.